# Capturing protein communities by structural proteomics in a thermophilic eukaryote

Panagiotis L Kastritis[1,†] (iD), Francis J O'Reilly[1,2,†] (iD), Thomas Bock[1,†] (iD), Yuanyue Li[1] (iD), Matt Z Rogon[1], Katarzyna Buczak[1], Natalie Romanov[1] (iD), Matthew J Betts[3], Khanh Huy Bui[1,4] (iD), Wim J Hagen[1], Marco L Hennrich[1] (iD), Marie-Therese Mackmull[1] (iD), Juri Rappsilber[2,5] (iD), Robert B Russell[3], Peer Bork[1], Martin Beck[1,*] (iD) & Anne-Claude Gavin[1,**] (iD)

## Abstract

The arrangement of proteins into complexes is a key organizational principle for many cellular functions. Although the topology of many complexes has been systematically analyzed in isolation, their molecular sociology *in situ* remains elusive. Here, we show that crude cellular extracts of a eukaryotic thermophile, *Chaetomium thermophilum*, retain basic principles of cellular organization. Using a structural proteomics approach, we simultaneously characterized the abundance, interactions, and structure of a third of the *C. thermophilum* proteome within these extracts. We identified 27 distinct protein communities that include 108 interconnected complexes, which dynamically associate with each other and functionally benefit from being in close proximity in the cell. Furthermore, we investigated the structure of fatty acid synthase within these extracts by cryoEM and this revealed multiple, flexible states of the enzyme in adaptation to its association with other complexes, thus exemplifying the need for *in situ* studies. As the components of the captured protein communities are known—at both the protein and complex levels—this study constitutes another step forward toward a molecular understanding of subcellular organization.

**Keywords** computational modeling; cryo-electron microscopy; fatty acid synthase; interaction proteomics; metabolon
**Subject Categories** Metabolism; Post-translational Modifications, Proteolysis & Proteomics; Structural Biology
**Mol Syst Biol.** (2017) 13: 936

## Introduction

As the molecular machines of the cell, protein complexes are the cornerstones of most biological processes, and are the smallest, basic functional and structural units of proteome organization (Duve, 1975; Gavin *et al*, 2002; Krogan *et al*, 2006). Many individual studies and extensive proteome-wide screens in a variety of organisms have identified comprehensive repertoires of protein complexes and have provided insights into their molecular composition and anatomy (Gavin *et al*, 2002; Krogan *et al*, 2006; Kuhner *et al*, 2009; Amlacher *et al*, 2011; Havugimana *et al*, 2012; Lapinaite *et al*, 2013; von Appen *et al*, 2015; Hoffmann *et al*, 2015; Wan *et al*, 2015; Yan *et al*, 2015). These studies relied on extensive biochemical purification, often including multiple sequential steps or dimensions, and so inherently selected for the most biophysically stable assemblies. However, protein complexes—as an organizational principle—cannot account alone for the complex integration of the many cellular processes *in situ*. Additional layers of functional organization, beyond free diffusion and random collision of functional biomolecules within organelles, are required to ensure, for example, the efficient transfer of substrates along enzymatic pathways (dubbed metabolons; Srere, 1987; Wu & Minteer, 2015; Wan *et al*, 2015; Wheeldon *et al*, 2016), the effective transduction of signals (Wu, 2013), and the synthesis of proteins according to the local cellular needs (Gupta *et al*, 2016). This requires spatially and temporally synchronized sets of protein complexes—protein communities (Barabasi & Oltvai, 2004; Menche *et al*, 2015)—which we define as higher-order, often dynamically associated, assemblies of multiple macromolecular complexes that benefit from their close proximity to each other in the cell. To date, protein communities have not been properly conceptualized because experimental frameworks to capture this higher-order proteome organization are missing.

---

1   European Molecular Biology Laboratory, Structural and Computational Biology Unit, Heidelberg, Germany
2   Chair of Bioanalytics, Institute of Biotechnology, Technische Universität Berlin, Berlin, Germany
3   Cell Networks, Bioquant & Biochemie Zentrum Heidelberg, Heidelberg University, Heidelberg, Germany
4   Department of Anatomy and Cell Biology, McGill University, Montreal, QC, Canada
5   Wellcome Trust Centre for Cell Biology, School of Biological Sciences, University of Edinburgh, Edinburgh, UK
    *Corresponding author. Tel: +49 6221 387 8267; E-mail: martin.beck@embl.de
    **Corresponding author. Tel: +49 6221 387 8816; E-mail: gavin@embl.de
    †These authors contributed equally to this work

We used cell fractions from a thermophilic eukaryote, *Chaetomium thermophilum* (Amlacher *et al*, 2011), to delineate and characterize protein communities in crude extracts that retain aspects of cellular complexity. Our experimental design, in particular our choice of a thermophilic organism to minimize the disassembly of protein–protein interactions and the respective fractionation conditions, favors the identification of especially higher molecular weight species. To cope with the complexity of such samples, we combined quantitative mass spectrometry (MS) with electron microscopy (EM) and computational modeling approaches. We computed a network capturing various communities and demonstrate its usefulness for further analysis. We used cross-linking mass spectrometry (XL-MS) and EM to validate our approach, which shows that crude cellular extracts retain the basic principles of proteome organization. They are amenable to high-resolution cryoEM analyses of the sociology of protein complexes within their higher-order assemblies. As the proteins can be readily identified within these extracts, our methodological framework complements the emerging single-cell structural biology approaches that provide high-resolution snapshots of subcellular features (Beck & Baumeister, 2016; Mahamid *et al*, 2016) but are currently unable to pinpoint the underlying biomolecular entities.

## Results

### Cellular fractions serve as a proxy for the cellular environment and retain basic principles of cellular organization

Many fundamental components of the cell were first structurally investigated from thermophilic archaea because protein interactions in thermophiles have higher stability compared to their mesophilic counterparts. We chose to study the thermophilic eukaryote, *Chaetomium thermophilum*, a promising model organism for structurally investigating eukaryotic cell biology, because protein communities may be more robust than those from other model systems.

Large-scale analyses based on extensive, multi-dimensional fractionation have been applied to characterize protein complexes from various organisms and cell lines. These have all demonstrated that protein complexes—as biochemically highly stable entities—are an ubiquitous organizational principle (Wan *et al*, 2015). Our goal here was to capture more transient, higher-order associations and to characterize the functional organization of a eukaryotic proteome under conditions that mimic the native, cellular state. To achieve this, we obtained simple and crude cellular fractions (simplified cell lysates) from the thermophilic fungus *C. thermophilum* by single-step analytical size exclusion chromatography (SEC; Fig 1). The chromatographic method used here achieves relatively high resolution compared with gel filtration methods commonly used on a preparative scale (Kristensen *et al*, 2012) and the resulting 30 fractions span molecular weights ranging from ~0.2 to ~5 MDa. We first analyzed these fractions in biological triplicate by label-free quantitative liquid chromatography–mass spectrometry (LC-MS/MS) to characterize co-eluting proteins, complexes, and communities. We identified 1,176 proteins across all fractions that were present in at least two of the triplicates (Dataset EV1, Appendix Fig S1A), which account for 27.4% of the expressed proteome of *C. thermophilum* (Bock *et al*, 2014). For comparison, in human HeLa and U2OS cell lines, 19 and 29% of the proteome elutes in these high molecular weight SEC fractions, respectively (Kristensen *et al*, 2012; Kirkwood *et al*, 2013). Of these 1,176 proteins, 97% have a molecular weight < 200 kDa as a monomer but were still reproducibly identified in fractions corresponding to larger molecular masses, suggesting that most are engaged in large macromolecular assemblies.

Next, we determined an experimental elution profile for each protein by quantifying protein abundance based on iBAQ scoring (Schaab *et al*, 2012). The abundance of the detected proteins spans five orders of magnitude (Appendix Fig S1B and C), demonstrating that relatively rare complexes are also captured in this process. The elution profiles correlate well across the biological triplicates (squared Pearson coefficient; $0.82 < r^2 < 0.88$; Appendix Fig S1B and C, and Dataset EV1). Similarly, the protein composition of each SEC fraction was generally highly reproducible (Pearson coefficient; $0.61 < r < 0.98$; Appendix Fig S1D and Dataset EV1). To further assess the quality and effectiveness of the biochemical separation, we determined whether the observed elution profiles matched the composition, molecular weight, and stoichiometries of well-characterized and conserved protein complexes as contained in the Protein Data Bank (PDB; Berman *et al*, 2000). We generated 3D interaction models for 378 out of the identified 1,176 *C. thermophilum* proteins using comparative structural modeling that takes into account species-specific differences (cutoffs: > 30% sequence coverage, > 30% sequence identity; Appendix Figs S2 and S3, Dataset EV2, details in the Materials and Methods). The resulting benchmark set of structurally known protein complexes comprises 34 heteromers (involving 212 proteins) and 166 homomers, the latter mainly consisting of metabolic enzymes (Appendix Fig S2E). As expected, the subunits of the heteromultimeric complexes typically co-eluted in the same biochemical fractions (Fig 2A, Dataset EV2 and Appendix Fig S4), although a considerable number of proteins showed multiple elution peaks indicating that they are engaged in various complexes (Kuhner *et al*, 2009). For 102 protein complexes that eluted in a single peak (Dataset EV2), we also compared their predicted molecular weights to those estimated from their retention time (tR) during SEC elution (Fig 2B). In 52 well-characterized cases —for example, the chaperonin-containing TCP-1 (CCT) complex or the 19S proteasome—we observed a good agreement between the expected and observed tRs, further validating the general efficiency of the cell lysate separation procedure. However, 50 protein complexes eluted at much higher molecular weights than anticipated from their structural models. These shifts are unlikely to be due to non-specific post-lysis protein aggregation as no visible aggregates were formed under our experimental conditions (EM analysis, see below). They are therefore probably functionally relevant as we observed that co-eluting complexes share the same functional ontology (independent two-sample *t*-test *P*-value = 3.88E-50, Appendix Fig S5) or directly interact (cross-linking experiments, see below), suggesting a functional relationship. This is consistent with the view that protein complexes might self-assemble with higher stoichiometries, contain additional components—*that is,* RNA, DNA, metabolites, or proteins—and/or form uncharacterized, protein communities. An interesting example is the glycolytic enzyme enolase (EC 4.2.1.11) that forms a structurally characterized dimer *in vitro* ($2 \times 47.7 = 95.4$ kDa; (Kuhnel & Luisi, 2001); PDB:2AL2) but seems to be part of a ~4-MDa assembly in the cellular fractions of *C. thermophilum* (Fig 2B). This supports previous

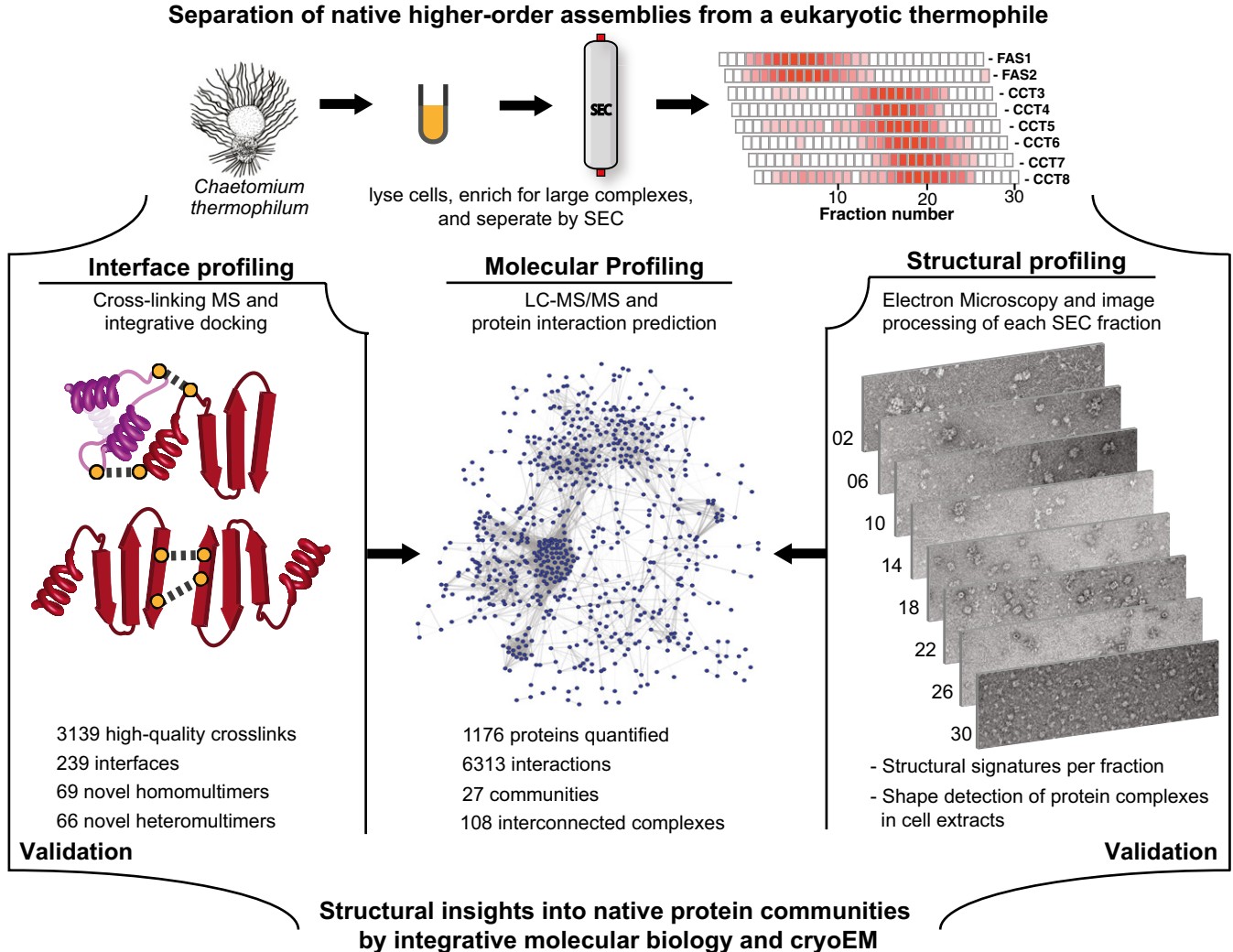

**Figure 1.  Overview of integrative structural network biology of native cell extracts in a thermophilic eukaryote.**

We combined computational modeling approaches adapted from network biology (molecular profiling) with molecular biophysics, electron microscopy (EM; structural profiling), and quantitative and cross-linking mass spectrometry (interface profiling) to systematically chart and characterize the organization of protein complexes into functional, local communities. Large-scale electron microscopy and cross-linking mass spectrometry are used as validation tools.

indications that enolase participates in higher-order multienzyme assemblies, such as the somewhat elusive eukaryotic glycolytic metabolon (Menard *et al*, 2014). Overall, our operational definition of protein communities using a reproducible and sensitive structural proteomics approach captures important snapshots of the functional organization of cellular proteomes.

**A compendium of *C. thermophilum* protein complexes within protein communities**

We next used the protein elution profiles in conjunction with known functional associations to systematically define protein communities. Correlations between profiles can indicate membership of the same complex (Havugimana *et al*, 2012; Kristensen *et al*, 2012) or of protein communities that perform functions in a spatiotemporal context. For all possible protein pairs in the dataset, we calculated a Pearson correlation coefficient (cross-correlation co-elution (CCC)

score), to measure the similarity of their elution profiles (see Materials and Methods for details). Although distinct complexes can share similar and overlapping elution profiles (Havugimana *et al*, 2012), CCC scores discriminate between random co-eluting and interacting protein pairs (Appendix Fig S6). To improve the assignment of interaction probabilities, we also exploited a set of indirect interactions (e.g. genetic interaction, colocalization) from the STRING database (v.9.1; Franceschini *et al*, 2013). These are based on orthologs from *Saccharomyces cerevisiae* (Dataset EV3) and a set of non-redundant structural interfaces that share homology with *C. thermophilum* predicted interfaces using Mechismo (Betts *et al*, 2015; Materials and Methods; Dataset EV3). We combined these two datasets with the interaction probabilities derived from the elution profiles. We used a random forest classifier trained with randomly sampled sets of true-positive ($N = 5,000$) and true-negative ($N = 5,000$) interactions that we extracted from public sources after manual curation (PDB (Berman *et al*, 2000) and affinity purification–mass

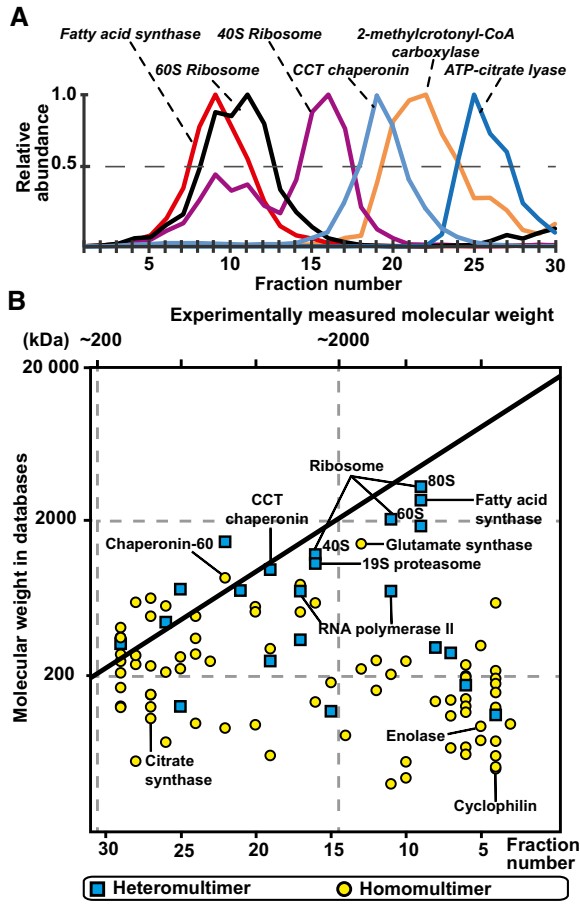

**Figure 2.  Identification of protein complexes and communities in the cellular extracts.**

A   Elution of selected protein complexes as a function of their retention times (see Appendix Fig S4 for their corresponding subunit elutions).

B   Scatter plot indicating discrepancies in the expected and measured molecular weights of 102 protein complexes that elute as a single peak; 50% of protein complexes are observed to have higher molecular weights than structurally characterized, indicating that they are organized in higher-order assemblies.

spectrometry (AP-MS) data (Benschop *et al*, 2010); Dataset EV3). We took a minimum interaction probability of 0.85 to construct a protein–protein interaction network (Appendix Figs S7–S10) that contains 679 proteins, 427 of which are not known to be members of protein complexes as their orthologs in yeast are not in any complex defined by PDB (Berman *et al*, 2000), AP-MS data (Benschop *et al*, 2010), or the Saccharomyces Genome Database (SGD; www.yeastgenome.org).

From this network, we used a clustering method that efficiently discovers densely connected overlapping regions that represent protein complexes and communities (ClusterONE; Nepusz *et al*, 2012). We systematized the recovery of protein complexes by an exhaustive parameter search and benchmarking (Sardiu *et al*, 2009) with the set of known structures (from the PDB) and yeast complexes (from AP-MS data; Dataset EV2; Materials and Methods). The optimal set of clustering parameters defines 21 clusters that

account for protein complexes and 27 clusters accounting for protein communities that contain 108 interconnected protein complexes as subsets (Fig 3). Importantly, varying the parameters had only marginal impact on the final protein content (Dataset EV3 and Materials and Methods), highlighting the robustness of the protein communities. Overall, the protein communities include 62% of the set of known protein complexes (the set of known PDB and AP-MS data, Dataset EV2) with 90% average coverage of their components (Fig 3 and Dataset EV4). Of these communities, a well-known example is the ribosome protein community, which comprises not only the stable 60S and 40S ribosomal complexes but also the translation initiation factor eIF2B that is only transiently associated with the ribosome (Fig 3, Appendix Fig S8A). Other examples are novel such as the physical interaction between the Tup1-Cyc8 corepressor and a histone deacetylase complex (community #22), which is consistent with recent functional data demonstrating that these two complexes indeed cooperate to robustly repress transcription in yeast (Fleming *et al*, 2014). The analysis also captured a lipid anabolism metabolon (community #23), which not only includes the homomultimeric complexes of a cytochrome b reductase (Cbr1, which regulates the catalysis of sterol by biosynthetic enzymes) and a choline-phosphate cytidylyltransferase (Pct1, which is a rate-determining enzyme of the CDP-choline pathway for phosphatidylcholine synthesis), but also several enzymes in the sterol synthesis pathway. The transmembrane protein suppressor of choline sensitivity 2 (Scs2) is also observed, which is a known regulator of phospholipid metabolism. Its presence may seem peculiar at first; however, this provides physical evidence for a role for this community in validating the interconnectivity of lipid and sterol metabolism in fungi (Parks & Casey, 1995). Such coordinated regulatory effects may functionally optimize membrane plasticity and specificity (Ramgopal & Bloch, 1983). This community presumably localizes at the endoplasmic reticulum (ER)–plasma membrane (PM) interface as this is thought to be the location of all five predicted transmembrane proteins (Dataset EV4).

The protein communities include associations that have been reported as transient, non-stoichiometric or of low abundance in other organisms. For example, the 19S regulatory particle of the proteasome was found to be associated with two known components, Upb6 and Nas6, and the 20S core particle with two mutually exclusive alternative cap proteins, Blm10 and Cdc48 (Kish-Trier & Hill, 2013; Fig 3, Appendix Fig S8A). The protein communities also capture transient interactions between nuclear transport receptors and transport channel nucleoporins—specifically, the interactions between karyopherins and the Nsp1 complex and the Nup159 complex (Appendix Fig S8B)—that have been elusive in standard biochemical experiments (Patel & Rexach, 2008) and that were recently found to have high off-rates (Milles *et al*, 2015). Elsewhere, RNA polymerase II is found in a community with several splicing complexes, the U2 snRNP, the U4/U6.U5 tri-snRNP, and the smD3 complexes (Appendix Fig S8B, Dataset EV4). These spliceosomal machineries are known to interact with RNA polymerase II via the carboxy-terminal domain of its largest subunit, ensuring the tight coupling of mRNA transcription and splicing (Martins *et al*, 2011). We thus consider that our approach successfully identifies higher-order associations of complex core modules.

This compendium of *C. thermophilum* protein communities (Dataset EV4), which are precisely assigned to specific and highly

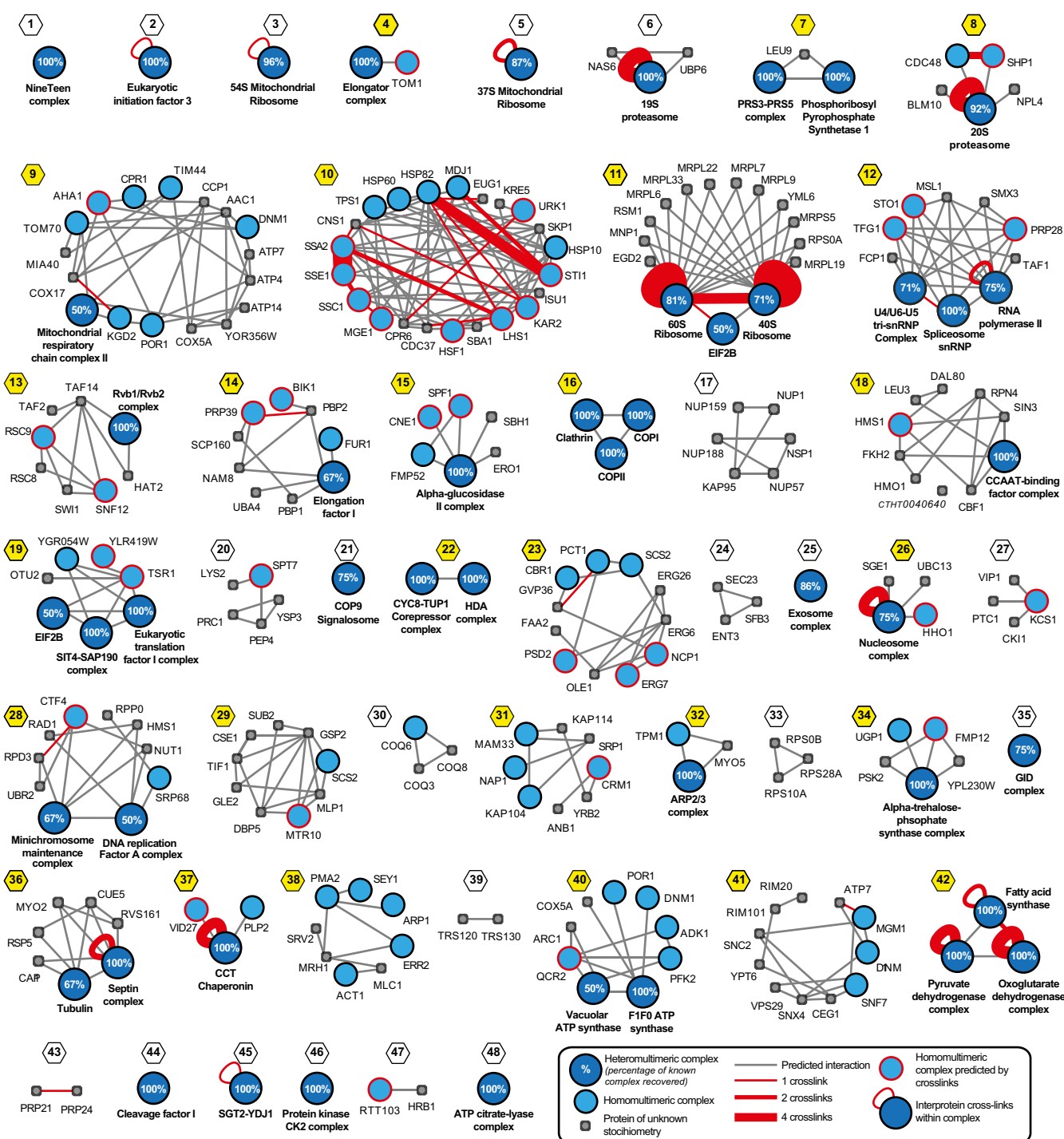

**Figure 3. Network derived from large-scale fractionation predicts 48 protein complexes and communities.**

Integration of experimental elution data, known functional associations, and predicted interaction interfaces from homologous proteins allow the creation of a high-quality network with interconnected protein complexes (Appendix Figs S8–S10). Here, known protein complexes are shown in blue and other physically associated proteins in gray, predicted interactions of complexes as gray lines and cross-links as red lines, and cross-links between different subunits of a heteromultimeric complex are represented with red loops (see insert). Communities containing multiple complexes are highlighted with yellow; numbering and naming of complexes and communities are described in the legend of Appendix Fig S9.

reproducible cellular fractions, represents an important resource for structural biologists (Appendix Fig S10). It not only captures transient associations but also identifies subunits of known complexes

that have so far remained elusive. Due to the evolutionary distance between *C. thermophilum* and most well-established model organisms, subunits of even highly conserved core complexes are not

necessarily identified (or unambiguously identified) by sequence alignments. As exemplified in Appendix Fig S8A, the co-elution data can be used to identify such subunits and to assign orthology (details in Dataset EV4) by narrowing down a set of protein complex member candidates based on their experimental profiles (e.g. Appendix Fig S8A).

## Characterization of new interaction interfaces by cross-linking mass spectrometry

Physical interactions inferred from co-occurrences can also be indirect, and so next we characterized the interaction interfaces occurring between members of the predicted protein communities by applying proteome-wide cross-linking MS (XL-MS) to the fractions (see Materials and Methods). To capture a large fraction of the interactome, we integrated three independent XL-MS datasets, which we acquired using different complementary protocols, for example, using different chemical cross-linkers, and both sequence-based and structure-based estimates of the false discovery rate (FDR; see Materials and Methods and Appendix Fig S11). We identified 3,139 high-quality cross-links (177 intermolecular and 2,962 intramolecular; Table 1) with sequence-based and structure-based FDRs of 10.0 and 12.0%, respectively (Dataset EV5). To validate the data, we checked which cross-linked peptide identifications are satisfied at the structural level, *that is,* correspond to distances between $C_a$ atoms of cross-linked lysine residues smaller than 33 Å ($Lys(c_a.c_a) < 33$ Å). A comparison with all structurally known complexes (see above and Dataset EV5) revealed that 73% of intermolecular and 84% of intramolecular cross-links were satisfied. In addition, the measured $Lys(c_a.c_a)$ distances effectively recapitulated the expected log-normal distributions covered by the disuccinimidyl suberate (DSS) and BS3 cross-linkers, which further validates the calibration method we employed (Fig 4A, Appendix Fig S11). A significant fraction of the cross-linked peptides ($N = 2,732$) mapped interactions within single polypeptide chains and therefore probably define intramolecular contacts (Dataset EV5). The remaining 407 cross-linked peptides define 118 heteromultimeric (177 cross-links) and 121 homomultimeric (230 cross-links) interfaces (Dataset EV5), which is largely consistent with our network analysis of protein communities (Fig 3) and the proteins forming the interconnected complexes (Appendix Fig S8A). Our analysis indicates that 135 (i.e. 56%) of these interfaces were previously unknown, and among the novel ones, 11 are between different complexes within the same community (Dataset EV5, Appendix Figs S11 and S12).

Overall, the cross-linking benchmarking methodology presented here suggests strict, but high-quality, structural validation that may be applicable to any cross-linking study on complex mixtures of proteins or complexes. For example, the XL-MS dataset validates a community of heat-shock complexes that elute with apparent molecular weights in the mega-dalton range (i.e. much higher than known complexes). We mapped nine new interfaces within this community, based on XL-MS data that suggest the existence of a complex interaction network or a chaperone community that comprises chaperones and co-chaperone complexes (Dataset EV5). Our XL-MS analysis further validates the notion of identifiable protein communities and is suggestive of several previously unknown interfaces that might be targeted for high-resolution structural studies.

## Characterization of structural signatures of protein communities from cell extracts using fatty acid synthase as an example

To demonstrate that crude cellular fractions are amenable to the structural characterization of protein communities, we examined the different fractions for recurring structural signatures using EM (Fig 4B–D) without adding any cross-linker for further stabilization of interactions. Specifically, we acquired a large set of negatively stained electron micrographs of all fractions, identified single particles, and subjected them to 2D classification. We used cross-correlation to identify structural signatures recurring across neighboring fractions and the number of single particles contained within a class as a proxy for abundance (see Materials and Methods for details). Several structural signatures were observed, some of which were clearly recognizable as corresponding to known protein complexes, for example, the fatty acid synthase (FAS), the proteasome, and the 40S and 60S ribosome (Fig 4C). In these cases, both the quantitative MS and EM data were highly consistent with the molecular weight and size of the given complexes (Fig 4B and C). These results confirm the high quality of our profiling data and illustrate how compositionally complex samples might be rapidly annotated on the structural level in the future. We also observed several potentially novel structural signatures using orthogonal biochemical separation (Fig 4D), demonstrating that a wealth of structural information can be mined with this approach.

We next analyzed one of these structural signatures—fungal FAS —in more detail. In our analysis, FAS is a structurally prominent, 2.6-MDa complex that contains six copies of all eight catalytic centers comprising the complete metabolic pathway for 16- and 18-carbon fatty acid production. It is known to functionally interact with various other enzymes (FAS1 and FAS2 have 16 high-confidence interactors in *S. cerevisiae* according to STRING). Consistent with this notion, additional electron optical densities, probably corresponding to associated protein complexes, are observed that sometimes form linear elongated arrangements (Fig 5A and Appendix Fig S13). The majority locate outside the reaction chambers of the central wheel that is clearly manifested in 2D class

**Table 1. Cross-linking statistics at a false discovery rate of 10%.**

| FDR 10% | Cross-links | Structurally mapped | Total interfaces covered | Novel interfaces |
|---|---|---|---|---|
| Total cross-links | 3,139 | 931 | 239 | 135 |
| Cross-links on monomers | 2,732 | 851 | – | – |
| Cross-links on homomultimers | 230 | 36[a] | 121 | 69 |
| Cross-links on heteromultimers | 177 | 44 | 118 | 66 |

[a]These cross-links show decrease in intra-residue distance when measured on known homomultimers by $26.3 \pm 13.4$ Å.

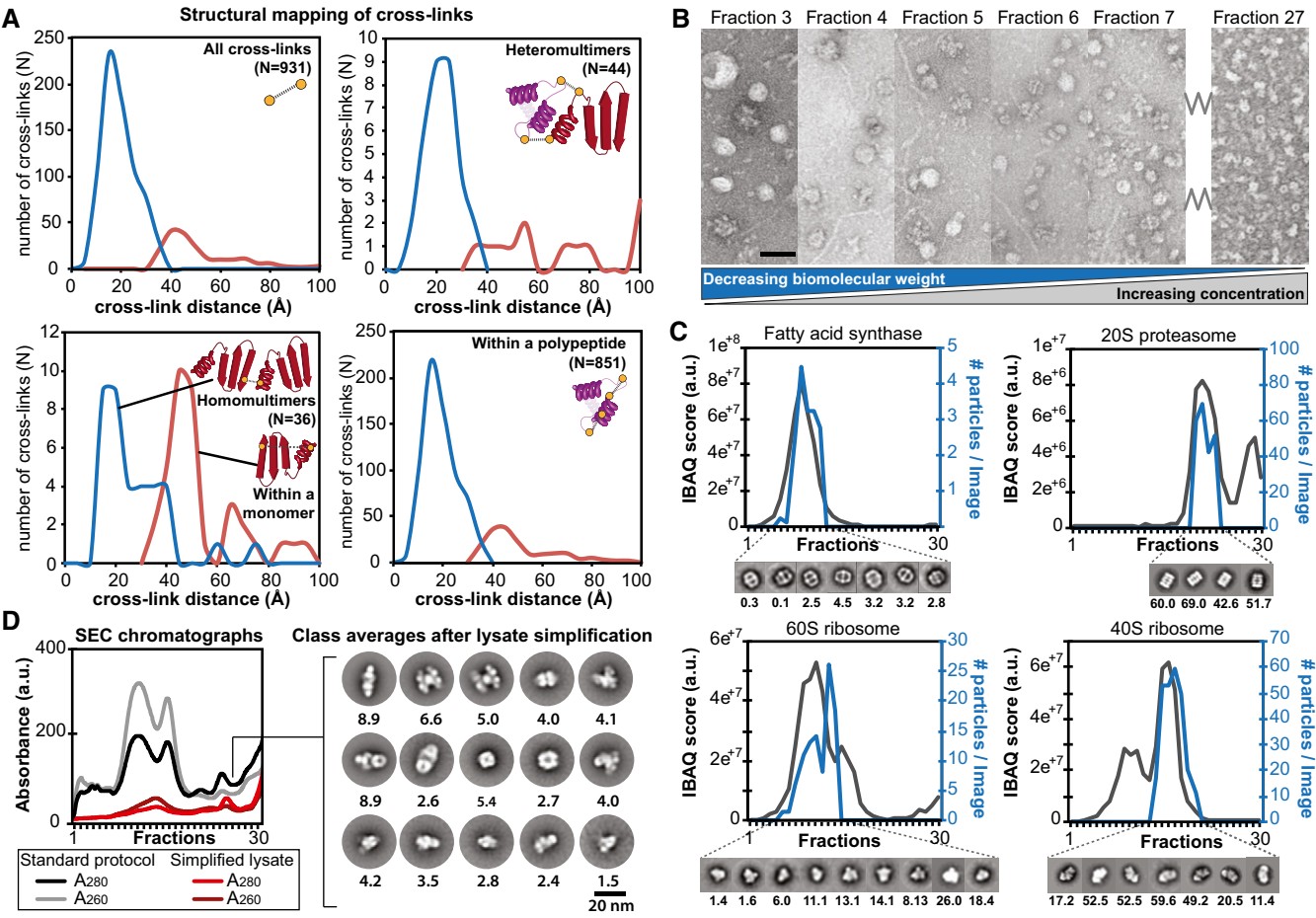

**Figure 4. Higher-order assemblies identified by proteome-wide cross-linking mass spectrometry, biomolecular modeling, and negative-stain electron microscopy.**

A  Distance distributions of identified cross-links on top of the modeled protein complexes and identification of novel interactions. Satisfied distances are shown in blue and over-length cross-links are shown in red.

B  Negatively stained electron micrographs of fractions 3–7 and 27 directly derived from size exclusion chromatography showing the structural signatures and their structural integrity within the fractions. Decreasing molecular weight correlates with increased protein concentration as a function of protein complex elution is highlighted. Scale bar: 60 nm.

C  Abundance profiles as determined by quantitative mass spectrometry correlate with the number of observed single particles of the corresponding structural signature within the negative-staining electron micrographs; shown for fatty acid synthase, 20S proteasome, 60S and 40S ribosome (the number of particles per image per fraction is indicated below the class averages).

D  Simplification of lysate (collecting only the flow-through from anion exchange chromatography) prior to SEC separation allows class averaging of structural signatures from complex fractions that were previously too low abundant. The number of particles per image per fraction is indicated below the class averages.

averages (Fig 5B and Appendix Fig S13). These additional electron optical densities proximal to FAS are seen more frequently than would be expected by random chance (Fig 5C). Their positioning at the entrance/exit tunnel of FAS (the malonyl transacylase domain) suggests the formation of a metabolon with other enzymes that deliver and accept substrates and products (as, for example, observed with acetyl-coA carboxylase (Acc1) in yeast using light microscopy; Suresh *et al*, 2015). To biochemically validate this observation, we utilized the fact that unlike FAS, many enzymes involved in fatty acid metabolism are covalently modified with the co-factor biotin. We therefore affinity-purified biotinylated proteins using avidin beads with subsequent XL-MS. The majority of the proteins in the eluate were known to be natively biotinylated except

for CTHT_0013320 (MCC2; the non-biotin-containing subunit of a carboxylase) and both subunits of FAS (CTHT_0037740, CTHT_0037750). We found the flexible acyl carrier protein (ACP) and malonyl/palmitoyl transferase (MPT) domains (that catalyze the first step in FA synthesis) to be cross-linked with the two sub-units of a carboxylase [CoA carboxylase beta-like; MCC2 and CTHT_0015140 (DUR1,2)] (Fig 5D). This interface characterized by cross-linking matches the one seen on the original cryoEM images and the 2D class averages and further supports the notion that a metabolon comprising other enzymes that deliver and accept substrates and products has been captured. Further corroborating the abovementioned findings, our SEC-MS co-elution data suggest an association of FAS with the same carboxylase (Dataset EV1). The

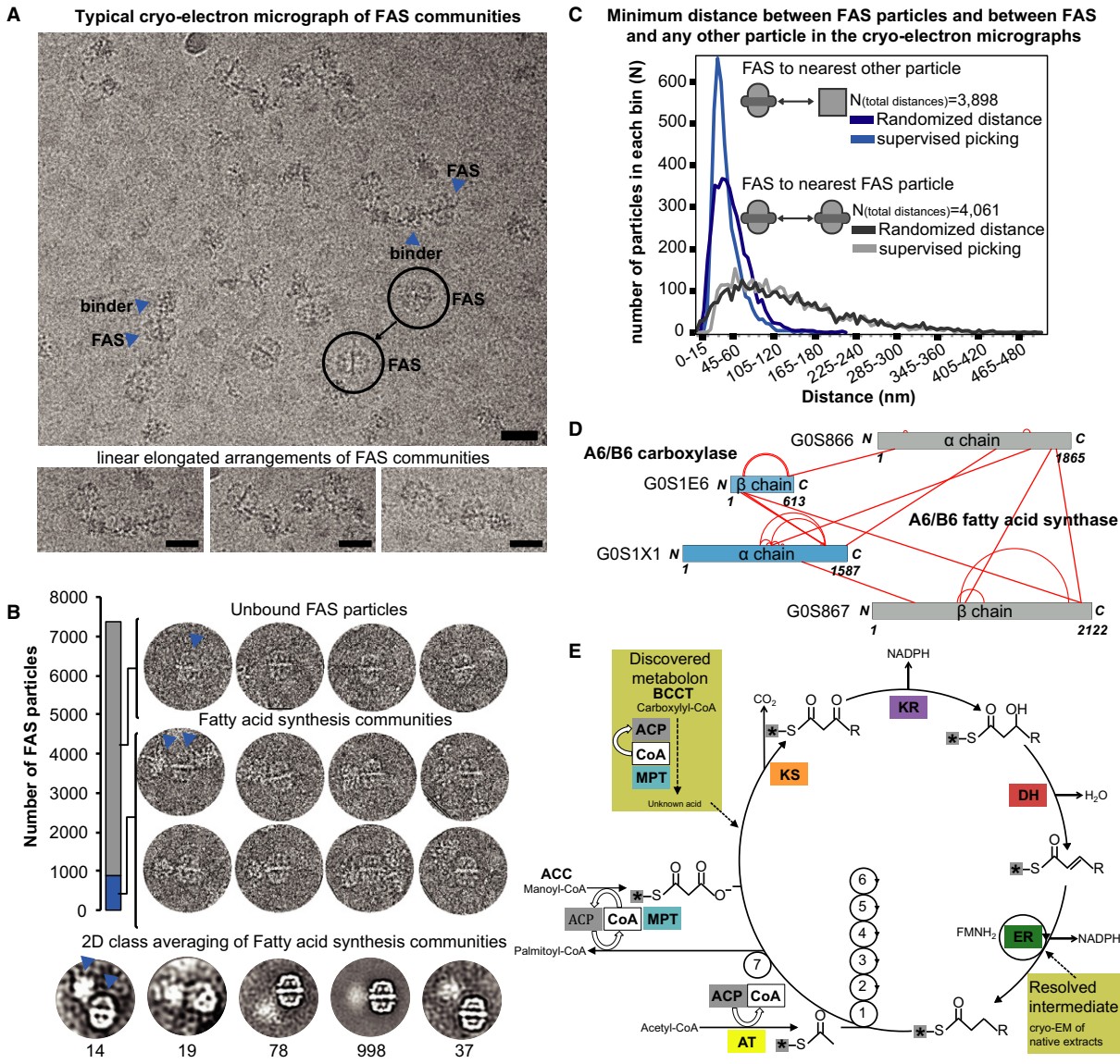

**Figure 5. Visualization of transient interactions in fatty acid synthesis.**

A   Communities in fatty acid metabolism and the quantification of intra-community distances within cryo-electron micrographs. Fatty acid synthase (FAS) frequently interacts with other sizeable protein complexes in a linear "pearl-string-like" arrangement and usually localizes at the edges of the community. Scale bars correspond to 25 nm. FAS particles (circles) and their nearest neighbors (arrow heads) are indicated.

B   Additional density outside of the ctFAS dome is observed in ~10% of the single particles; 2D class averages shown at the bottom. The arrow heads show typical assemblies within the pool of particles. In 90% of all cases, isolated FAS particles are seen (unbound state). In the remaining 10%, higher-order protein assemblies comprised of FAS particles and high molecular weight binders are seen (bound state).

C   Related to (A). Calculation of minimum distances between pairs of FAS molecules as well as FAS molecules and their closest non-FAS neighbors in comparison with random distributions. Whereas FAS molecules are randomly distributed, their binders are not, confirming physical interactions. Supervised picking means that all single particles were manually picked from the images. Randomized distance means that these manually picked particles were assigned random coordinates in each image (randomization of *x, y* coordinates considering image borders) and then their distance is calculated.

D   Cross-linking mass spectrometry data show that the binder is a carboxylase that is bound to the malonyl transacylase domain and acyl carrier protein (ACP) is in the vicinity, considering cross-link length and the positions of the lysine on the ctFAS structure. Cross-links come from both affinity-purified and fractionated cell extracts.

E   The molecular mechanisms in fatty acid synthesis (Wakil *et al*, 1983), and the relevance of the position of the ACP (see Fig 6 for details) and carboxylase to the catalytic cycle is indicated (see text). ACP, acyl carrier protein; CoA, acetyl-coenzyme A; MPT, malonyl/palmitoyl transferase; KS, ketoacyl synthase; KR, ketoacyl reductase; DH, dehydratase; ER, enoyl reductase; AT, acetyltransferase. Asterisks represent the acyl carrier protein (ACP).

organization of these domains in close proximity to each other implies a mechanism of substrate delivery from the carboxylase to FAS (Fig 5E). It is likely that this could be an alternate substrate for

either odd-chain fatty acid synthesis (Fulco, 1983) or, less likely, fatty acid branching (Kolattukudy *et al*, 1987), provided via direct substrate channeling (Fig 5E). FAS and the carboxylase are known

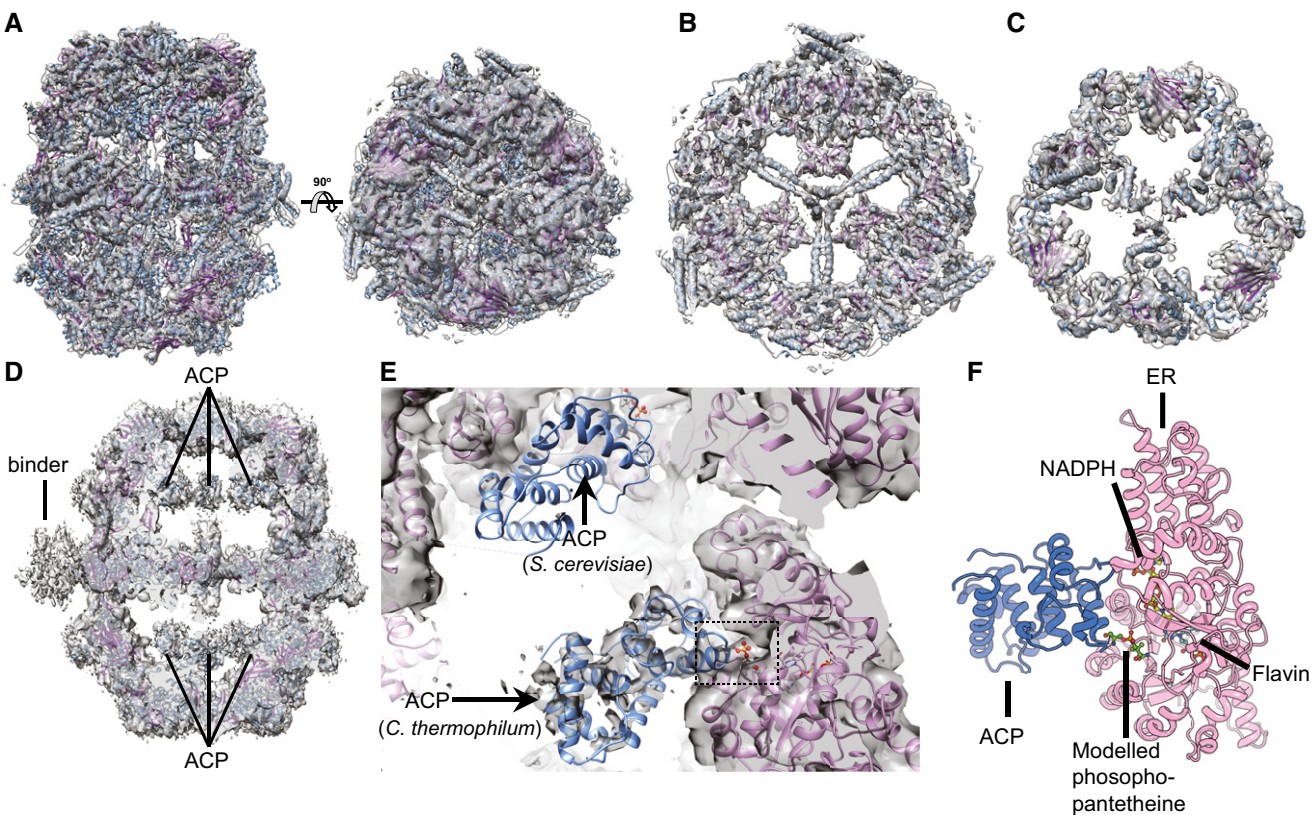

**Figure 6. CryoEM structure of fatty acid synthase resolved to 4.7 Å as obtained from cryo-electron micrographs of fraction numbers 7–9.**

A   The cryoEM map of *Chaetomium thermophilum* fatty acid synthase (ctFAS) is shown isosurface rendered and superimposed with the fitted X-ray structure of yeast FAS (Jenni *et al*, 2007). The domes and the cap show the unambiguous fit of α-helices and β-strands.

B   A slice through the central wheel of fungal FAS. The pitch of α-helices is resolved.

C   As for (B) but sliced through the dome structure.

D   Location of acyl carrier protein (ACP) within the cryoEM map of ctFAS and the position of additional density outside the dome.

E   Fit of ACP in the cryoEM map of ctFAS and comparison with the crystallographically determined location in yeast FAS; additional density within the active site possibly resembling the acyl chain bound on the ACP is observed.

F   Molecular model of the interaction between the ACP and the ER domains of ctFAS in cartoon representation. The model was derived from a rigid fit from (B) and subsequently flexibly refined for clash removal and interface energetics optimization.

to be two independent complexes and would therefore fit our definition of a community.

We next set out to test whether high-resolution structure determination is possible in these crude extracts. A high-resolution structure of FAS in isolation has been determined by X-ray crystallography (Jenni *et al*, 2007; Leibundgut *et al*, 2007; Lomakin *et al*, 2007). Using cryoEM (Gipson *et al*, 2010; Boehringer *et al*, 2013), certain regions—in particular the lid—remained unresolved, probably due to intrinsic flexibility. We acquired 1,917 cryo-electron micrographs of the relevant biochemical fraction and identified 7,370 single particles displaying the relevant structural signature in 1,597 micrographs (~83%). Structural analysis and 3D classification resulted in a reconstruction at ~4.7 Å containing only 3,933 particles (Appendix Fig S14), demonstrating that high-resolution structural analysis by cryoEM is feasible in complex cellular fractions. Overall, the cryoEM map of ctFAS recapitulated the X-ray structure of fungal FAS relatively well (Fig 6A–C), including high-resolution details such as the helical pitch in the central wheel (Fig 6B, Appendix Fig S15). In contrast to previous cryoEM structures of fungal FAS, even

the lid region was clearly resolved (Fig 6C, Appendix Fig S15). Thermophilic proteins are more susceptible to structural analysis by X-ray crystallography and NMR because they contain less flexible loops (Amlacher *et al*, 2011; Lapinaite *et al*, 2013). Our data indicate that this also extends to cryoEM, possibly because of reduced flexibility. Strikingly, the cryoEM structure did exhibit additional low-resolution density outside the reaction chambers that probably corresponds to the community discussed above (Fig 6D). Further, the ACP that iteratively shuttles the substrate within the catalytic chamber of FAS (Jenni *et al*, 2007) was captured at a different active site, albeit at slightly reduced resolution (Fig 6E and Appendix Fig S15). In previous structures, ACP located near the ketoacyl synthase domain involved in the first step in fatty acid synthesis (Jenni *et al*, 2007). Here, ACP is located in the vicinity of the enoyl reductase (ER; Fig 6E and F, and Appendix Fig S15) that reduces the α-β-double-bond of the acyl chain to a single bond. This final catalytic step in acyl chain metabolism is targeted by important antibacterial and antifungal drugs (e.g. Triclosan and Triclocarban, Atromentin and Leucomelone).

# Discussion

The hypothesis of an intermediate layer of molecular sociology between supramolecular assemblies and organelles (Srere, 1987; Wu & Minteer, 2015) states that protein complexes spatially and temporally co-exist and directly interact with each other or individual proteins to form higher-order assemblies within specific cellular compartments, referred to here as protein communities (see Box 1). Such communities would be capable of channeling substrates for efficiency, could regulate pathway flux by transient binding kinetics, and would be formed by higher-order interactions (e.g. macromolecular crowding, excluded volume effects, "stickiness" of the cytoplasm, hydrodynamic interactions; Srere, 1987) and are attractive targets for biotechnology to increase reaction efficiencies (Wheeldon *et al*, 2016). Until now, a direct visualization or comprehensive analysis of such complexity was missing.

Although cellular fractions are more similar to the cellular environment than highly purified samples, they are less so than vitreous sections of the true cellular environment that nowadays can be studied using cryo-electron tomography but not by MS methods. We have shown that it is possible to capture at least some aspects of these protein communities in a systematic way using an integrative structural biology approach on cell fractions of the eukaryotic thermophile *C. thermophilum,* a model organism for structural biology as its proteins exhibit superior biochemical stability (Amlacher *et al*, 2011; Lin *et al*, 2016). The fractionation of cell extracts was postulated to retain close to native cellular interactions decades ago (Mowbray & Moses, 1976), but due to the molecular heterogeneity of these extracts, it was long thought to be prohibitive to structural characterization. In this study, we demonstrate that cellular fractions preserve basic principles of proteome organization and enable the identification of protein communities that are directly amenable to high-resolution cryoEM analyses. As a case in point for the latter, we structurally captured a specific catalytic step in fatty acid synthesis as well as some of the interfaces between FAS and other molecules using cryoEM in this setting. A wealth of other recurring structural signatures was identified, some readily recognizable but others novel and requiring further molecular characterization—a promising finding for structural and molecular biologists.

Overall, we designed an integrative approach specifically designed to identify and structurally characterize higher-order biomolecular assemblies. The specific elements implied the use of a chromatographic column to separate high molecular weight cellular assemblies and the choice of a thermophilic organism (to minimize the disassembly of protein–protein interactions upon lysis). The follow-up analyses are also tuned to cope with the large size of the communities (i.e. XL-MS with a cross-linker that identifies interactions up to 3 nm distance, and EM methods which are advantageous for higher-order assemblies of large molecular weight). The method described here is dedicated to the identification of protein communities, although of course other biomolecules such as nucleic acids or lipids might be part of the identified communities and contribute to their association. The combination with other identification strategies such as RNA sequencing and small molecule MS might further enlighten this aspect in the future. The broader applicability of cryoEM to non-purified samples will be limited by the abundance and the stability of the

---

**Box 1.  Organization models of a community involving enzymatic pathways.**

*Definition*: Protein communities are higher-order, often dynamically associated, assemblies of multiple macromolecular complexes that benefit from their close proximity to each other in the cell. Protein communities imply spatially and temporally synchronized sets of protein complexes. They ensure, for example, the efficient transfer of substrates along enzymatic pathways (dubbed metabolons and illustrated in the bottom panel), the effective transduction of signals, and the synthesis of proteins according to the local cellular needs. The concept goes beyond the classical linear representations of pathways that imply freely diffusing and randomly colliding biomolecules (bottom right panel). The assembly of protein communities sometimes require molecular scaffolds (e.g. RNA, biological membranes, or structural proteins), and can be regulated by post-translational modifications. Shapes 1–4 in the panel below show different enzymes of a sequential pathway. In contrast to free diffusion, these enzymes might also multimerize within a community to increase reaction efficiencies. *Methods*: The characterization of protein communities implies their retrieval from *in vivo*, cellular and physiological contexts, and the choice of a thermophilic organism is expected to minimize their disassembly. Their biochemical purification can be achieved via affinity purification coupled to mass spectrometry (Gavin *et al*, 2002; Huttlin *et al*, 2017), or, more efficiently, directly in crude cellular fractions that retain aspects of cellular complexity and favor the identification of especially higher molecular weight species (this study). The latter is also amenable to the systematic characterization of protein communities through integrative structural biology approaches, implying for example, quantitative cross-linking mass spectrometry (XL-MS), electron microscopy (EM), and molecular, biophysical modeling.

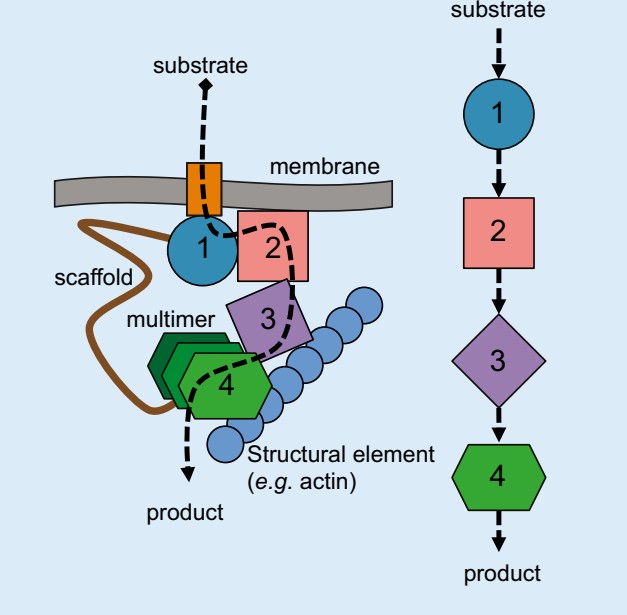

---

protein communities during the lysis procedure. However, methods to improve the stability of these interactions, potentially with cross-linking prior to fractionation or lysis, would allow discovery of further dynamic interactions and protein communities, and would allow further simplification of the protein mixtures for structural study using this pipeline. Advances in EM acquisition and data analysis methods might further improve the coverage and identification of protein communities in the future.

The emerging *in cellulo* structural biology approaches, based on the electron tomographic analyses of entire cells, have already started to produce the next generation of "big data" (Beck & Baumeister, 2016). These approaches hold great potential to structurally define protein communities in their native environment, the cell. They however fall short in the biochemical and molecular identification of these communities, as single-cell mass spectrometry is likely to remain limited to the few most abundant proteins for the near future. We anticipate that our approach that targets crude cellular extracts of intermediate molecular complexity as a proxy for the cellular milieu will crucially complement *in cellulo* methods because it allows a direct correlation between structural and molecular signatures.

# Materials and Methods

### Separation of *C. thermophilum* communities

*Chaetomium thermophilum* communities were enriched from cell lysates by spin filtration and fractionated using a Biosep SEC-S4000 (7.8 × 600) size exclusion chromatography (SEC) column from Phenomenex, Germany (see Appendix Supplementary Methods).

### Protein co-elution prediction and mass spectrometry

Protein abundances were recorded from each SEC fraction by liquid chromatography–mass spectrometry (LC-MS). Prediction of protein co-elution was performed by Pearson correlation of protein abundance profiles. LC-MS data were processed using the MaxQuant (Cox & Mann, 2008).

### Cross-linking/mass spectrometry

The cross-linking datasets searched with xQuest: Isotope-coded disuccinimidyl suberate (DSS; Creative Molecules) was used to perform cross-linking reactions as described previously (Walzthoeni *et al*, 2012). Cross-linked peptides were enriched by gel filtration before LC-MS analysis. All LC-MS data were obtained from an Orbitrap Velos Pro instrument (Thermo Scientific). Search and FDR were performed with the xQuest/xProphet (Leitner *et al*, 2014) software. For the cross-linking dataset searched with Xi, samples were cross-linked using a 1:1 w:w ratio of protein to BS3 (Thermo Scientific). Cross-linked peptides were enriched by gel filtration before LC-MS analysis. All LC-MS data were obtained from an Orbitrap Fusion Lumos Tribrid mass spectrometer (Thermo Scientific). Search and FDR were performed with the Xi (Giese *et al*, 2016) and XiFDR (Fischer & Rappsilber, 2017) software suites.

### Prediction of protein communities

For each protein pair, interactions based on structural homologs were predicted using Mechismo (Betts *et al*, 2015), *Saccharomyces cerevisiae* orthologs were found using eggNOG (Jensen *et al*, 2008), and interaction data (excluding physical interactions) were downloaded from String (v.9.1; Franceschini *et al*, 2013). These data were combined with co-elution data from the SEC analysis using a Random Forest (RF) classifier and a manually curated training set of

reference interactions to filter out spurious connections and infer a network of high-confidence predicted interactions. Protein complexes and communities were inferred using ClusterONE (Nepusz *et al*, 2012). The cross-linking ld score (Walzthoeni *et al*, 2012) was calibrated on distance restraints imposed by the cross-linker. Cross-linking distances were calculated by Xwalk (Kahraman *et al*, 2011) using structural models.

### Structure prediction of proteins participating in high molecular weight assemblies

Prediction of the structure of all 1,176 identified proteins was performed with iTASSER v4.2 (Yang *et al*, 2015) and Modeller 9v2 (Sali & Blundell, 1993). The best predicted model was selected according to its respective c-score (Roy *et al*, 2010). Details for model quality (for those with > 30% of sequence identity and coverage) are shown in Appendix Fig S2.

### Protein complex assignment using Protein Data Bank and calibration of cross-linking quality

Each of the 1,176 proteins found in total in all three biological replicates was submitted to the NCBI BLAST server (http://blast.ncbi.nlm.nih.gov/Blast.cgi) and searched against the Protein Data Bank (PDB; www.pdb.org). A threshold of 30% of sequence identity was assigned. A decision on the assembly was taken after back-BLASTing the rest of the subunits, if any, of the PDB structure to the *C. thermophilum* proteome. All results are included in Dataset EV2.

### Modeling of protein interfaces using cross-linking data

HADDOCK was used for modeling protein interfaces (de Vries *et al*, 2010; van Zundert *et al*, 2016). Cross-linking data were implemented as interaction restraints, set to have an effective (and maximum) $C\alpha$-$C\alpha$ distance of 35.2 Å, whereas the minimum distance was defined only by energetics.

### Negative-stain electron microscopy and 2D class averaging

Samples were directly deposited on glow-discharged (60 s) Quantifoil®, type 300 mesh grids and negative-stained with uranyl acetate 2% (w/w) water. Recording of data was performed with a side-mounted 1K CCD Camera (SIS). After data acquisition (pixel size = 7.1 Å), E2BOXER was used for particle picking (37,424 particles were picked out of 30 fractions). Class averaging was performed using RELION 1.2 (Scheres, 2012a,b). Cross-correlation of final class averages was performed using MATLAB v7.4.

### *ct*FAS enzyme preparation and vitrification for cryoEM

ctFAS was ~50% enriched (see Appendix Fig S14) and overall protein concentration was determined to be ~40 ng/μl. Samples were then deposited on glow-discharged (60 sec) carbon-coated holey grids from Quantifoil®, type R2/1. A FEI Vitrobot® was used for plunge-freezing. In short, humidity was set to 70%, blotting and drain time to 3 and 0.5 s, respectively. Sample volume applied was 3 μl and blot offset was set to −3 mm.

## CryoEM image acquisition, data processing, and 3D reconstruction

The vitrified samples were recorded on a FEI Titan Krios microscope at 300 kV. Pixel size was set to 2.16 Å and a FEI Falcon 2 camera was used in movie mode. Total dose applied was summed to 48 $e^-/$Å$^2$, but the last frame was used only for particle picking. A total number of 13,419 micrographs were acquired in 21 h (1 frame/6 s; 1 movie/42 s). Motion correction was applied to acquired micrographs (Li *et al*, 2013). E2BOXER was used for particle picking. CTFFIND was used for CTF correction (Rohou & Grigorieff, 2015). The RELION 1.2 package (Scheres, 2012a,b) was then used for 2D class averaging, 3D classification, and 3D reconstruction of the density map. Default Gaussian mask from RELION 1.2 gave a calculated resolution (gold standard FSC = 0.143) of 4.7 Å.

## Modeling of the ACP–enoyl reductase domain interaction and the FAS–carboxylase metabolon

Models of *C. thermophilum* acyl carrier protein (ACP) and enoyl reductase (ER) domains were generated using Modeller 9v2 and chosen structural homologs were selected from the yeast homolog with resolved densities for both (Leibundgut *et al*, 2007). Additional density of ACP was observed close to the ER domain of fatty acid synthase (FAS); thus, coarse placement of the ACP was performed using CHIMERA (Pettersen *et al*, 2004) and subsequently fitted to the density. Energy calculations were performed as previously described (Kastritis & Bonvin, 2010; Kastritis *et al*, 2014). Correlation of van der Waals energy with experimentally measured equilibrium dissociation constants for known complexes is derived from Kastritis *et al* (2014).

## Data and software availability

The primary datasets produced in this study are available in the following databases:
Structural data: EMDB EMD-3757 http://www.ebi.ac.uk/pdbe/entry/emdb/EMD-3757.
Proteomics data: PRIDE PXD006660 http://www.ebi.ac.uk/pride/archive/projects/PXD006660.
Cross-linking data: PRIDE PXD006626 http://www.ebi.ac.uk/pride/archive/projects/PXD006626.
List of identified proteins: Dataset EV1.
List of protein complexes: Dataset EV2.
Results of network benchmarking: Dataset EV3.
List of protein communities: Dataset EV4.
Results from the cross-linking experiments: Dataset EV5.
Cytoscape file: Dataset EV6.

**Expanded View** for this article is available online.

## Acknowledgements

PLK and TB acknowledge Marie Curie Actions for the EMBL Interdisciplinary Postdoc (EIPOD) fellowship. The authors acknowledge Vera van Noort (KU Leuven) for kindly providing the orthologous genes in yeast for interaction mapping. The EMBL Electron Microscopy and Proteomics Core Facilities are acknowledged. The authors thank the Gavin and Beck laboratory members for valuable discussions. We thank all groups and group leaders in the EMBL Structural and Computational Biology Unit for inspiring discussions and creating a stimulating and vibrant environment. Part of this work was supported by the Wellcome Trust through a Senior Research Fellowship to JR [grant number 103139]. The Wellcome Centre for Cell Biology is supported by core funding from the Wellcome Trust [grant number 203149]. This work was supported by CellNetworks (Excellence Initiative of the University of Heidelberg) with funds given to RBR, PB, MB and A-CG.

## Author contributions

A-CG, MB, PB, RBR, and JR supervised and administered the project and secured funding. A-CG, MB, PB, PLK, RBR, FJO'R, and TB wrote the manuscript. PLK designed and performed the electron microscopy experiments, carried out computation and modeling, analyzed the data, wrote the code, and solved the structure. FJO'R and TB designed and performed the SEC and the quantitative MS/MS and XL-MS experiments. MZR, FJO'R and PLK designed and analyzed the network. KHB and WJH contributed to cryoEM data acquisition and analysis. NR computationally analyzed data. YL, KB, M-TM, and MLH contributed to the SEC and MS experiments. MJB and RBR contributed the homologous interfaces for the network. All authors read and approved the final manuscript.

## Conflict of interest

The authors declare that they have no conflict of interest.

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
