## [Review Process File · Molecular Systems Biology]

Manuscript EMBO-2016-7412

Capturing protein communities by structural proteomics in a thermophilic eukaryote

Panagiotis L. Kastritis, Francis J. O'Reilly, Thomas Bock, Yuanyue Li, Matt Z. Rogon, Katarzyna Buczak, Natalie Romanov, Matthew J. Betts, Khanh Huy Bui, Wim J. Hagen, Marco L. Hennrich, Marie-Therese Mackmull, Juri Rappsilber, Robert B. Russell, Peer Bork, Martin Beck & Anne-Claude Gavin

Corresponding authors:

Martin Beck & Anne-Claude Gavin, European Molecular Biology Laboratory

Review timeline:

Submission date:	26 October 2016
Editorial Decision:	15 December 2016
Revision received:	13 April 2017
Editorial Decision:	26 May 2017
Revision received:	12 June 2017
Accepted:	20 June 2017

Editor: Thomas Lemberger

Transaction Report:

1st Editorial Decision

15 December 2016

The major issues raised by the reviewers refer to the following points:

- while all reviewers, and the editor, see the integrated cryoEM/MS approach as one of the major promising strengths of the study, it seems that an extension to the (primary) characterization and identification of (some) additional complexes would go a very long way towards a more convincing demonstration of the potential of this approach to systematically investigate higher order protein assemblies.
- a more careful discussion is needed to examine rigorously the issues related to interaction specificity and to avoid overinterpretation of the available evidence such as GO-term enrichment.
- the concept of 'protein communities' should be better defined.
- reviewer #3 also mentions the absence of functional validation.

Given the balance of opinions expressed by the reviewers, we feel that we can ask for a major revision of the present work. While the issue of functional validation is relevant, we feel that expanding on the cryoEM/MS structural proteomics approach would in fact be more important,

On a more editorial level, we would also kindly ask you to include a "Data availability" section that

provide a list of the resource name, accession number and resolvable link to the datasets presented in this work.

- As you may have noticed, we recently replaced Supplementary Information by Expanded View (EV, see examples in <http://msb.embopress.org/content/11/6/812>). In this format, a limited number of Supplementary Figures (max 5) can be integrated in the article as EV figures that are interactively collapsible/expandable and will be typeset by the publisher. In this case, the figures should be cited as "Figure EV1, Figure EV2" etc... in the text and their respective legends should be added to the main text after the legends of regular figures. The illustrations should be provided as separate files.

- For the figures that you do NOT wish to display as Expanded View figures items, they should be bundled together with their legends in a 'traditional' supplementary PDF, now called the **Appendix**. Appendix should start with a short Table of Content and the figures should be named and referred to in the main text as: "Appendix Figure S1, Appendix Figure S2" etc. See detailed instructions regarding expanded view here: <http://msb.embopress.org/authorguide#expandedview>.

- Additional Tables/Datasets should be labeled and referred to as Table (or Dataset) EV1 etc. Table/Dataset legends can be provided in a separate tab in case of .xls files. Alternatively, you can upload a .zip file containing the Table/Dataset file and a separate README .txt file with the legend/description.

- We would also encourage you to include the source data for figure panels that show essential data, so that readers can download these data directly from the figure. Source data files are associated to individual panels of main figures. **Numerical data** should be provided as individual .xls files (including a tab describing the data) or csv or tab-delimited text files. **For 'blots' or microscopy**, uncropped images should be submitted. For **network visualization**, Cytoscape session files, if available, can be supplied. The files should be labeled as "Source Data for Figure 1A" etc. Source Data for Expanded View and Appendix figures should be uploaded as a single ZIP file containing all the Source Data for Expanded View and Appendix content. (Additional information on source data is available in the "Guide for Authors" section at <http://msb.embopress.org/authorguide#sourcedata>).

REFEREE REPORTS

Reviewer #1:

The manuscript by Kastritis et al. attempts the characterization of a level of intracellular organization beyond protein complexes, that of protein communities. This level involves functional and structural interactions between different protein complexes. Crude cell extracts are analysed by the combination of SEC-MS, XL-MS, and cryoEM, and the data are integrated with available structural models and protein-protein interaction data. The analysis results in what the authors describe as a compendium of 48 protein communities.

The study contains sufficient elements of conceptual novelty and the aim is relevant. Generating a map of high-order protein communities in a cell would constitute a step forward in our understanding of intracellular organization and provide a useful resource for further biological studies. The integrative structural biology approach proposed by the authors is interesting and could constitute a new powerful strategy for the large-scale analysis of protein complexes in situ, a current challenge in structural biology. Another element of novelty is the high-resolution cryoEM analysis of SEC fractions, without the need to obtain biochemically homogenous samples.

Despite its potential impact, the manuscript suffers from some conceptual and technical weaknesses, which in my view prevent publication in its present form.

Major points:

1. It is not clear how the authors define the novel concept of protein communities. Consequently, it is difficult to evaluate the extent to which the aim of mapping protein communities on a large scale

is reached. Based on the Introduction, protein communities are "spatially and temporally synchronized sets of protein complexes" or "higher-order, often dynamically associated, assemblies of multiple macromolecular complexes that benefit from their close proximity to each other in the cell." This definition requires the participation of multiple (minimum two) protein complexes in the same community. However, less than half of the 48 protein communities identified by the authors seem to fulfil this criterion based on the representation in Fig. 3 (e.g., communities: 7-12, 14-16). The other communities include single, previously characterized complexes (e.g., communities 3 or 5) or interactions of known complexes with other proteins, which could be interpreted as potentially new subunits of the complex (e.g., community 1). The second and the third groups would be of limited interest since they would not pertain to the higher level of organization the authors wish to map. What is the source of this discrepancy? Should the definition of communities be broader than that presented in the Introduction (i.e., any cluster resulting from their clustering approach, ideally supported by prior data)? Did the authors identify only 22 protein communities based on their definition? Or are all the grey nodes shown in Fig. 3 themselves members of a known protein complex? The definition of community must be clarified and how the different types of interactions shown in Fig. 3 meet this definition must be explained.

2. Based on Fig. 1 it appears that the authors used an integrated approach involving SEC, structural models, XL-MS, and EM to map out protein communities on a large scale. In fact, the identification of protein communities relied only on the correlation of SEC-MS elution profiles and prior data supporting the interaction. This should be made clear in the text and figures.

EM is only used as a follow-up tool to further characterize one interaction from one of the identified communities, the interaction of FAS with another complex. Surprisingly, XL-MS also does not significantly contribute to the mapping of communities. The coverage of the XL-MS analysis appears quite low. 300 intermolecular crosslinks with a 5% FDR were recently reported by another group based on analysis of a whole human cell extract (Liu et.al Nat. Meth, 2015, 12:1179). In the study submitted to MSB, the authors identify on average 1-2 intermolecular crosslinks per SEC fraction, accounting to a total of 52 identified intermolecular crosslinks in the *Chaetomium* extract at a 7% FDR. Given the significantly lower complexity of SEC fractions compared to a whole cell extract, I would expect many more relevant crosslinks to be identified. Further, only four of the identified crosslinks (Fig. 3) seem to support the identification of a protein community - that is, the crosslink is between proteins known to belong to two different protein complexes (e.g., the crosslink in community 42). The others link different monomeric proteins or a known complex to new proteins. Based on this, the identification of interfaces between different complexes from the same community seems a weak point.

Thus, XL-MS and EM are only used as validation tools for specific cases, which reduces the impact of the proposed methodological framework. The authors should clarify the role of EM and XL-MS in their pipeline (for example by adapting Figure 1) and specify that they are not exploited for a large-scale analysis. They should also explain that large-scale XL-MS and/or EM data would strengthen the identification of protein communities, which currently relies only on SEC and prior data. Further, a discussion on the technical limitations of the XL dataset should be added.

Minor points

1. SEC-MS coupled to protein correlation profiling has already been applied by the Lamond, Marcotte and other groups to identify members of protein complexes. An element of novelty of this manuscript is the specific focus on the identification of higher molecular weight assemblies. I suggest that the authors stress the elements of their experimental pipeline that were specifically designed to support identification of these higher molecular weight species (e.g., SEC conditions to specifically separate HMW protein assemblies, choice of a thermophilic organism to minimize the disassembly of protein-protein interactions upon lysis, etc.).

2. Related to the previous point, the authors should explain at the beginning of the Results or in the Introduction their choice of focusing on *Chaetomium thermophilum*. Now the reader wonders until the Discussion why *C. thermophilum* was chosen. The formation of protein communities could be a specific property of this organism. A comment should be added to the Discussion.

3. For each dark-blue node (heteromultimeric complexes) in Fig. 3, how many subunits were identified in the SEC-MS community out of how many total known subunits? This information should be added to the figure.
4. How does the structural follow-up of FAS relate to the data shown in Fig. 3, community 42? Are any of the interactions shown in Fig. 3 confirmed or further characterized?
5. Fig. 5, Panel C: the authors should clarify in the legend what they mean with 'randomized distance' and 'supervised picking'.
6. Fig. 4, Panel A: The annotation outside the boxplots is not clear. The authors could re-organize the presentation of these data

Reviewer #2:

General comments: In their manuscript "Capturing protein communities by structural proteomics in a thermophilic eukaryote," Kastiris et al employ a variety of biochemical, computational, and structural techniques to separate, evaluate, and identify protein components of a variety of larger "protein communities" in various regions of a cell. While a number of other groups have attempted analyses to achieve similar ends, the processes employed in this report are both clever and largely effective. This manuscript represents a useful step forward in the area of understanding the structural organization of protein communities within a eukaryotic cell, and I recommend acceptance of this manuscript with minor revisions.

Specific comments:

1. I found the manuscript difficult to follow in many places. One way to improve clarity is to make figure legends less terse and explain the results of the analysis more clearly for readers interested in the subject of protein complexes but not familiar with the lingo used in the manuscript.
2. Introduction, first sentence - the authors state that the crude extracts mimic the native cellular state. While these are more similar to the cellular environment than highly purified samples, they are less so than might occur with vitreous sections of the true cellular environment. Indeed, the process of lysis and fractionation likely separates each sample from some neighbors and poorly bound non-protein components. While the approach is laudable, and has offered improvements in terms of the ability to identify components compared to true in situ analyses (as stated at the end of that paragraph), the authors should consider revising this statement to reflect the middle-ground that such samples represent.
3. While protein-protein associations are of critical interest, many such protein communities also associate with non-protein molecules, such as RNA and DNA, as well as carbohydrate, lipid, and other components. Moreover, the clarification step of their lysis ("Lysate was cleared of debris by centrifugation at 100,000 g for 45 min") suggests that they may be removing many organellar or nucleotide-associated communities. Were non-protein components identified in the mass spec step? Can the authors speak to the potential bias of their system, and possible mechanisms for analyzing these missing complexes or components?
4. In the main text on page 4, the authors state that their mass spec analysis covers 27.4% of the expressed proteome, while on page 5, they state that their data (which data?) recapitulates (sic) 62% of known protein complexes, with 70% of their known components. Can these numbers be broken down to analyze what types of components (small? Large? Membrane-associated? Post-translationally modified?) are missing, and how these various percentages relate? What is the source of "known complexes" in this case? Is this the network analysis? (the statement "see above" is unclear) The authors have used a variety of sources, combined in various ways, to validate their communities, so a defined list or source for what the final validation dataset of "known complexes" is would be useful. Could the authors reference the precise workbook page in their supplementary material, if this is already provided there? If not, could it be added?

5. Which samples were used for negative stain and cryo-EM analysis? Were these cross-linked or not? If they were cross-linked, it would be useful to also include a micrograph of non-cross-linked fractions as a control.

6. A final note about the discussion of the EM analysis - the authors acknowledge that they specifically examined their fractions for "structural signatures", essentially searching for obvious, very large, easily identifiable stable complexes. While their ability to derive a de novo structure of ctFAS is admirable, the broader applicability of this to non-purified samples is doubtful, given that the majority of stable complexes will be significantly smaller. Some discussion of these types of problems would be useful.

Minor points:

1. Figure S1 - header on panel B has the word "across" misspelled
2. Text page 5 (see note 3 above) - the word data is plural, so the "data recapitulate"

Reviewer #3:

The authors have separated lysates from the fungus, *Chaetomium*, into crude fractions, which they have analyzed in detail with mass spectrometry and cryoelectron microscopy to determine the abundance, interactions and structure of their constituent proteins and complexes. They present evidence that these constituents form dynamically interconnected "communities", which are often dissociated by more rigorous subcellular fractionation methods but have been preserved in these crude fractions. In a striking application of state-of-the art electron microscopy, the authors could identify and structurally characterize a single complex and present evidence for a previously structurally unobserved interacting partner. Despite this, there are a number of issues with the work as currently presented.

The idea of "protein communities" is not at all new, indeed it has been around since the inception of the interactome concept (which has thoroughly connected PPIs for which no stable complex has yet been observed), and it has been well understood for some time that our current "stable complex-centric" view of the interactome is incomplete (e.g. Menche, J. et al. Disease networks. Uncovering disease-disease relationships through the incomplete interactome. *Science* 347, 1257601 (2015)). Indeed, many of these concepts underlie the basics of subcellular fractionation (e.g. De Duve, C. Exploring cells with a centrifuge. *Science* 189, 186-94 (1975)). How does this "communities" concept differ from or significantly modify or contribute to the already well-established scale-free models of protein interactions in modern interactomics? (e.g. Barabási, A.-L. & Oltvai, Z. N. Network biology: understanding the cell's functional organization. *Nat Rev Genet* 5, 101-113 (2004)).

While the EM work presented is extremely impressive, on page 4 it is claimed that a failure to observe aggregates by EM is evidence of a lack of non-specific / post-lysis effects on the protein interactions observed. This seems an unreasonable notion. Firstly, any non-specific interactions will be distributed across a continuum of possible states of complexation and structures (reflecting the lack of specificity and probable lack of physiology), and therefore will not be likely to present coherent features permitting the grouping and classification needed by the EM methods used. The claim that functional ontology is a secondary validation for this method, also seems to be misplaced - in a simple example, groups of nuclear proteins may be more likely to interact non-specifically with other nuclear proteins than cytoplasmic proteins, post-lysis, i.e. despite lack of any physiological significance. Or, any protein may be more likely to interact with related proteins, metabolically speaking, once the cell is broken, although those proteins may normally be spatially separated and never make a contact in vivo. Suffice it to say, while GO enrichment has its uses and can permit hypothesis formation regarding the potential relationship between groups of proteins (which would be an appropriate way to present the observed GO enrichment here), it does not constitute validation of specificity or physiology. The crosslinking results do not provide a more compelling argument either, since they will trap what is present, post lysis, at sufficient concentration - bona fide or not - and, overall coverage appears to be low (probably due to sample complexity).

Fatty acid synthase also represents one of the most abundant complexes by far in yeasts, its subunits ranking ~20th most abundant (top 5%) of all proteins (<http://pax-db.org/> and references therein). It is not particularly surprising, then, that this could be identified and characterized in a crude fraction. This segment would have been much more impressive if a significant number of less abundant complexes (exosomes, exocysts, populations of different coated vesicles, etc.) had been separately identified and classified, at least at a primary level.

The potential to support the metabolon hypothesis is exciting, as it is well known that supposed metabolic complexes are extremely difficult to co-purify despite knowledge of functional interactions and co-localization data by microscopy. The method presented most likely does not constitute one with innate qualities that improve the likelihood of observing metabolons from a sample preparation perspective, but rather, leveraged the sensitivity of the MS instruments used in conjunction with standard fractionation and crosslinking to find a nugget to weakly support the potential power of the method. Perhaps if the paper were organized in a metabolon-centric manner (with more examples) it would be clearer how substantial and significant the enrichment of otherwise difficult to capture, intact metabolic complexes is.

Much time is spent describing the need for in situ analytical means but the methods employed here are anything but, and it is unclear if these crude fractions really do present a major advantage over advanced subfractionation and purification methods using e.g. crosslinking and careful optimizations to preserve weakly-interacting partners, or true in situ high resolution cryoEM tomography. If the authors want to make this claim they should devote time to rationalizing or at least explaining their protein extraction and preparation choices. Why that chosen extraction solution? For example, both EDTA and Mg⁺⁺ are used (and EDTA is listed twice) - why? Secondly, the handling time is significant compared to many modern approaches - 45 min of centrifugation (at a very high speed compared to typical extract preparation of ~20k RCF - probably to avoid slow protein concentration by ultrafiltration - and then the concentration step (how long?). By the time the sample is ready for loading on the SEC column or otherwise carried to the next experimental manipulation, a lifetime has passed (in terms of molecular half-lives in any random in vitro solution) - further making the claims of lack of non-specificity to be heavy-handed. In any case, the centrifugation used probably pellets stable, giant non-specific aggregates (which is indeed analytically important) - but one must ask about the accounting: what percentage of every protein complex highlighted in this study was found to be partitioned in the supernatant as opposed to the pellet? What is the partitioning bias of the communities described? At a bare minimum the authors should do some A/B/C comparisons of different protein extraction conditions, and/or extracts in the same condition prepared at different concentrations, in order to lend weight to their claims of what is happening in vitro and what observations are robust and reliable. Overlapping those data with established in vivo / in situ data (and ignoring high throughput IPs, or at least +/- such data) might reveal which condition is most useful for maintaining the in situ milieu, and therefore most suitable for accurate characterization by all the other methods employed.

Finally, and very significantly - there's simply no proper validation in this study. If the conclusions are valid - and probably some of them are - then it should be easy enough to select a hand full of nodes that represent interacting community members, and via deletion or mutation of those nodes, observe a phenotype and/or alter (subtly or significantly) the fractionation / MS profiles - this study has led the authors to hypothesize a particular behavior and arrangement of protein complexes/communities, and now they should test that claim by orthogonal experimentation to the path that led them there (and draw our attention to numerous interactions found in this study that resolve long standing paradoxes concerning functional associations that could not be shown to be physically linked previously). Which unexpected, un-described, or counter-intuitive nodes have been revealed here and contribute meaningfully to a physiological cascade they have not previously been (functionally) drawn into? Of course phenotype detection can be challenging - but at least some candidates must be validated in vivo to solidify the conclusions. I question that anything about this study can be characterized as in situ and recommend against that phrasing.

We would like to thank the reviewers for their overall positive comments and very constructive critique. We agree that our definition of protein communities required more clarity. We have

carefully revised the main text to address this issue and made a very clear distinction between data that contributed to the modeling of communities (quantitative MS, genetic interactions, predicted interfaces and computational clustering) and validation approaches (cross-linking mass spectrometry (XL-MS) and electron microscopy (EM)).

We have addressed the concerns regarding the comprehensiveness of the XL-MS data and the lack of additional validation. For this purpose, we collaborated with Juri Rappsilber's laboratory and made an effort to integrate complementary XL-MS methods into our workflow. As a result, acquired one of the most comprehensive data sets of complex samples to date. The number of cross-links that validate the identified communities has been considerably improved.

Reviewer #1:

The manuscript by Kastriitis et al. attempts the characterization of a level of intracellular organization beyond protein complexes, that of protein communities. This level involves functional and structural interactions between different protein complexes. Crude cell extracts are analyzed by the combination of SEC-MS, XL-MS, and cryoEM, and the data are integrated with available structural models and protein-protein interaction data. The analysis results in what the authors describe as a compendium of 48 protein communities.

The study contains sufficient elements of conceptual novelty and the aim is relevant. Generating a map of high-order protein communities in a cell would constitute a step forward in our understanding of intracellular organization and provide a useful resource for further biological studies. The integrative structural biology approach proposed by the authors is interesting and could constitute a new powerful strategy for the large-scale analysis of protein complexes in situ, a current challenge in structural biology. Another element of novelty is the high-resolution cryoEM analysis of SEC fractions, without the need to obtain biochemically homogenous samples.

We would like to thank the reviewer for these positive comments.

Despite its potential impact, the manuscript suffers from some conceptual and technical weaknesses, which in my view prevent publication in its present form.

Major points:

1. It is not clear how the authors define the novel concept of protein communities. Consequently, it is difficult to evaluate the extent to which the aim of mapping protein communities on a large scale is reached. Based on the Introduction, protein communities are "spatially and temporally synchronized sets of protein complexes" or "higher-order, often dynamically associated, assemblies of multiple macromolecular complexes that benefit from their close proximity to each other in the cell." This definition requires the participation of multiple (minimum two) protein complexes in the same community. However, less than half of the 48 protein communities identified by the authors seem to fulfil this criterion based on the representation in Fig. 3 (e.g., communities: 7-12, 14-16). The other communities include single, previously characterized complexes (e.g., communities 3 or 5) or interactions of known complexes with other proteins, which could be interpreted as potentially new subunits of the complex (e.g., community 1). The second and the third groups would be of limited interest since they would not pertain to the higher level of organization the authors wish to map. What is the source of this discrepancy? Should the definition of communities be broader than that presented in the Introduction (i.e., any cluster resulting from their clustering approach, ideally supported by prior data)? Did the authors identify only 22 protein communities based on their definition? Or are all the grey nodes shown in Fig. 3 themselves members of a known protein complex? The definition of community must be clarified and how the different types of interactions shown in Fig. 3 meet this definition must be explained.

We agree that the definition of protein communities coined in the introduction was only partially consistent with the results part in which we referred to them rather as the outcome of our clustering approach.

We have therefore revised the result section, on page 5, last paragraph as follows: "*From this network, we used a clustering method that efficiently discovers densely connected overlapping*

regions that represent protein complexes and communities (ClusterONE (Nepusz et al, 2012)). We systematized the recovery of protein complexes by an exhaustive parameter search and benchmarking (Sardiu et al, 2009) with the set of known structures (from the PDB) and yeast complexes (from AP-MS data) (Tables S2; Materials and Methods). The optimal set of clustering parameters defines 21 clusters that account for protein complexes and 27 clusters accounting for protein communities that contain 108 interconnected protein complexes as subsets (Fig. 3). Importantly, varying the parameters had only marginal impact on the final protein content (Table S3 and Materials and Methods), highlighting the robustness of the protein communities. Overall, the protein communities include 62% of the set of known protein complexes (the set of known PDB and AP-MS data, Table S2) with 90% average coverage of their components (Fig. 3 and Table S4). We also modified the Figure 3 (and Figure S8A), which follows this simplified and unambiguous definition.”

Further, we have removed the example Figure panel 4B, which includes docking of Mgm1p with ATP synthase, because it did not correspond to a community, but it was rather a protein complex with an additional subunit.

We also added in the second paragraph of the introduction, on page 3:

“We used cell fractions from a thermophilic eukaryote, Chaetomium thermophilum (Amlacher et al, 2011), to delineate and characterize protein communities in crude extracts that retain aspects of cellular complexity. Our experimental design, in particular our choice of a thermophilic organism to minimize the disassembly of protein-protein interactions and the respective fractionation conditions, favor the identification of especially higher molecular weight species. To cope with the complexity of such samples, we combined quantitative and cross-linking mass spectrometry (MS) with electron microscopy (EM) and computational modeling approaches. We computed a network capturing various communities and demonstrate its usefulness for further functional analysis. We used cross-linking mass spectrometry (MS) and electron microscopy (EM) to validate our approach, which shows that crude cellular extracts retain the basic principles of proteome organization. They are amenable to high-resolution cryoEM analyses of the sociology of protein complexes within their higher-order assemblies. As the proteins can be readily identified within these extracts, our methodological framework complements the emerging single-cell structural biology approaches that provide high-resolution snapshots of subcellular features (Beck & Baumeister, 2016; Mahamid et al, 2016) but are currently unable to pinpoint the underlying biomolecular entities.”

2. Based on Fig. 1 it appears that the authors used an integrated approach involving SEC, structural models, XL-MS, and EM to map out protein communities on a large scale. In fact, the identification of protein communities relied only on the correlation of SEC-MS elution profiles and prior data supporting the interaction. This should be made clear in the text and figures. EM is only used as a follow-up tool to further characterize one interaction from one of the identified communities, the interaction of FAS with another complex.

We followed this reviewer suggestion and clarified the fact that protein communities were identified using SEC-MS and prior data from structural and functional experiments. The XL-MS and EM data were used as follow-up and validation experiments, which brought additional experimental evidence for the existence of such communities. Fig. 1 is now revised to highlight this fact. We also point this out in the main text on page 3, second paragraph:

“We used cross-linking mass spectrometry (MS) and electron microscopy (EM) to validate our approach, which shows that crude cellular extracts retain the basic principles of proteome organization. They are amenable to high-resolution cryoEM analyses of the sociology of protein complexes within their higher-order assemblies. As the proteins can be readily identified within these extracts, our methodological framework complements the emerging single-cell structural biology approaches that provide high-resolution snapshots of subcellular features (Beck & Baumeister, 2016; Mahamid et al, 2016) but are currently unable to pinpoint the underlying biomolecular entities.”

Surprisingly, XL-MS also does not significantly contribute to the mapping of communities. The coverage of the XL-MS analysis appears quite low. 300 intermolecular cross-links with a 5% FDR were recently reported by another group based on analysis of a whole human cell extract (Liu et al. Nat. Meth, 2015, 12:1179). In the study submitted to MSB, the authors identify on average 1-2 intermolecular cross-links per SEC fraction, accounting to a total of 52 identified intermolecular cross-links in the Chaetomium extract at a 7% FDR. Given the significantly lower complexity of

SEC fractions compared to a whole cell extract, I would expect many more relevant cross-links to be identified. Further, only four of the identified cross-links (Fig. 3) seem to support the identification of a protein community - that is, the cross-link is between proteins known to belong to two different protein complexes (e.g., the cross-link in community 42). The others link different monomeric proteins or a known complex to new proteins. Based on this, the identification of interfaces between different complexes from the same community seems a weak point.

To address the reviewer's point, we have made an effort to scale up this part of the manuscript during the revision. XL-MS is still a challenging approach. Indeed, because of the kinetics and efficiency of the cross-linking reactions, it remains difficult to sample complex interactomes significantly with a single experiment or protocol. To circumvent this limitation and capture a larger fraction of the interactome, we developed additional cross-linking protocols that were used to complement the previous dataset. We thus collected additional XL-MS datasets of SEC fractions: (a) One at EMBL with an Orbitrap Velos Pro and subsequent analysis with xQuest and xProphet, which is more extensive in terms of peptide fractionation and the number of MS runs as compared to the data set included into the original version of the manuscript. (b) Another one in collaboration with the Rappalber laboratory in Berlin using complementary methods - that is a different cross-linker, BS3, a more sensitive instrument (Orbitrap Fusion Lumos Tribrid) and their in-house search engine Xi.

Together the new data make one of the biggest cross-linking datasets recorded with **3,139** cross-linked peptide pairs at 10% FDR (target-decoy) including **177** interlinks. Indeed, if we account for the fact we cross-link a much smaller number of proteins (i.e. 1176 proteins that eluted at high molecular weights) compared to those present in complete cellular lysates, our data compare favorably with other recent studies:

-isolated mitochondria, 2,427 peptide pairs, 578 inter-links, 5% FDR (Schweppe et al, PNAS, 2017, 144:1737) [PMID 28130547]

-extracts from whole human cells, 2,426 peptide pairs, 413 inter-links, 5% FDR, Liu et al Nat. Meth, 2015, 12:1179 PMID 26414014]

We used this information to validate a larger fraction of our data communities (see revised Figure 3). Overall, the XL-MS data cover 135 new interfaces, among them 35 are between different complexes from within the same community.

Thus, XL-MS and EM are only used as validation tools for specific cases, which reduces the impact of the proposed methodological framework. The authors should clarify the role of EM and XL-MS in their pipeline (for example by adapting Figure 1) and specify that they are not exploited for a large-scale analysis. They should also explain that large-scale XL-MS and/or EM data would strengthen the identification of protein communities, which currently relies only on SEC and prior data. Further, a discussion on the technical limitations of the XL dataset should be added.

As mentioned above, we have rephrased the introduction to make the contributions of the individual methods more clear (page 3) and we have improved the coverage of our XL-MS analysis page 7. We have also revised Figure 1 and its legend to highlight this point. We included the following statement in the discussion page 10: *"The broader applicability of cryoEM to non-purified samples will be limited by the abundance and the stability of the protein communities during the lysis procedure. However, methods to improve the stability of these interactions, potentially with cross-linking prior to fractionation or lysis would allow discovery of further dynamic interactions and protein communities, and would allow further simplification of the protein mixtures for structural study using this pipeline. Further advances in EM acquisition and data analysis methods might further improve the coverage and identification of protein communities in the future."*

Minor points

1. SEC-MS coupled to protein correlation profiling has already been applied by the Lamond, Marcotte and other groups to identify members of protein complexes. An element of novelty of this manuscript is the specific focus on the identification of higher molecular weight assemblies. I suggest that the authors stress the elements of their experimental pipeline that were specifically designed to support identification of these higher molecular weight species (e.g., SEC conditions to

specifically separate HMW protein assemblies, choice of a thermophilic organism to minimize the disassembly of protein-protein interactions upon lysis, etc.).

This was a very good suggestion. As mentioned above we rephrased the introduction (page 3) as follows: *“We used cell fractions from a thermophilic eukaryote, Chaetomium thermophilum (Amlacher et al, 2011) to delineate and characterize protein communities in crude extracts that retain aspects of cellular complexity. Our experimental design, in particular our choice of a thermophilic organism to minimize the disassembly of protein-protein interactions and the respective fractionation conditions, favor the identification of especially higher molecular weight species.”*

We also included the following statement into the discussion page 10:

“Overall, we designed an integrative approach specifically designed to identify and structurally characterize higher-order biomolecular assemblies. The specific elements implied the use of a chromatographic column to separate high molecular weight cellular assemblies and the choice of a thermophilic organism (to minimize the disassembly of protein-protein interactions upon lysis). The follow up analyses are also tuned to cope with the large size of the communities (i.e. XL-MS with a cross-linker that identifies interactions up to 3 nm distance, and EM methods which are advantageous for higher-order assemblies of large molecular weight).”

2. Related to the previous point, the authors should explain at the beginning of the Results or in the Introduction their choice of focusing on Chaetomium thermophilum. Now the reader wonders until the Discussion why C. thermophilum was chosen. The formation of protein communities could be a specific property of this organism. A comment should be added to the Discussion.

We added an explanatory section in the beginning of the results part (page 3) regarding C. thermophilum:

“Many fundamental components of the cell were first structurally investigated from thermophilic archaea because protein interactions in thermophiles have higher stability compared to their mesophilic counterparts. We chose to study the thermophilic eukaryote, Chaetomium thermophilum, a promising model organism for structurally investigating eukaryotic cell biology, because protein communities may be more robust than those from other model systems.”

3. For each dark-blue node (heteromultimeric complexes) in Fig. 3, how many subunits were identified in the SEC-MS community out of how many total known subunits? This information should be added to the figure.

This information has been added to Fig. 3. Most complexes are fully covered, and coverage spans from 50% to 100%.

4. How does the structural follow-up of FAS relate to the data shown in Fig. 3, community 42? Are any of the interactions shown in Fig. 3 confirmed or further characterized?

The cryoEM analysis of FAS suggests the presence of several associated protein complexes, with heterogeneous shapes that sometimes even form linear elongated arrangements (Supplementary Fig. 5A and S13) and supports the view that FAS is involved in several communities (as Fig. 3 does). The one with the carboxylase was not captured by our clustering approach, possibly because carboxylase also occurs as highly abundant independent species.

Many interactions shown in Fig. 3 are now confirmed and further characterized. Indeed, the improved coverage of the XL-MS analysis now leads to 11 validated novel inter-complex interactions communities for which we could model new interfaces, e.g. interactions involving heat shock complexes see text page 7.

5. Fig. 5, Panel C: the authors should clarify in the legend what they mean with 'randomized distance' and 'supervised picking'.

The terminology was clarified in the Figure legend. We now additionally wrote:

“Supervised picking means that all single-particles were manually picked from the images. Randomized distance means that these manually picked particles were assigned random coordinates

in each image (randomization of x,y coordinates considering image borders) and then their distance is calculated.”

6. Fig. 4, Panel A: The annotation outside the boxplots is not clear. The authors could re-organize the presentation of these data

The presentation of the data in Panel A was changed. Now, the cross-linking data are presented in a new Table in the main manuscript (Table 1) and the distances are plotted with an in-plot cartoon explanation, describing what has been measured for each.

Reviewer #2:

General comments: In their manuscript "Capturing protein communities by structural proteomics in a thermophilic eukaryote," Kastiris et al employ a variety of biochemical, computational, and structural techniques to separate, evaluate, and identify protein components of a variety of larger "protein communities" in various regions of a cell. While a number of other groups have attempted analyses to achieve similar ends, the processes employed in this report are both clever and largely effective. This manuscript represents a useful step forward in the area of understanding the structural organization of protein communities within a eukaryotic cell, and I recommend acceptance of this manuscript with minor revisions.

We would like to thank the reviewer these positive comments.

Specific comments:

1. I found the manuscript difficult to follow in many places. One way to improve clarity is to make figure legends less terse and explain the results of the analysis more clearly for readers interested in the subject of protein complexes but not familiar with the lingo used in the manuscript.

We made an effort to revise the main text and legends to explain our findings better to a broad readership. All changes have been highlighted in grey in the main text.

2. Introduction, first sentence - the authors state that the crude extracts mimic the native cellular state. While these are more similar to the cellular environment than highly purified samples, they are less so than might occur with vitreous sections of the true cellular environment. Indeed, the process of lysis and fractionation likely separates each sample from some neighbors and poorly bound non-protein components. While the approach is laudable, and has offered improvements in terms of the ability to identify components compared to true in situ analyses (as stated at the end of that paragraph), the authors should consider revising this statement to reflect the middle-ground that such samples represent.

To address this point, we included the following sentence into the Discussion on page 9: *“Although cellular fractions are more similar to the cellular environment than highly purified samples, they are less so than vitreous sections of the true cellular environment that nowadays can be studied using cryo electron tomography but not by MS methods.”*

3. While protein-protein associations are of critical interest, many such protein communities also associate with non-protein molecules, such as RNA and DNA, as well as carbohydrate, lipid, and other components. Moreover, the clarification step of their lysis ("Lysate was cleared of debris by centrifugation at 100,000 g for 45 min") suggests that they may be removing many organellar or nucleotide-associated communities. Were non-protein components identified in the mass spec step? Can the authors speak to the potential bias of their system, and possible mechanisms for analyzing these missing complexes or components?

Indeed, non-proteinaceous material is known to modulate or scaffold protein interactions, but they are not identified by our method. To make this clearer we included the following passage into the Discussion on page 10:

“The method described here is dedicated to the identification of protein communities, although of course other biomolecules such as nucleic acids or lipids might be part of the identified communities and contribute to their association. The combination with other identification strategies such as RNA sequencing and small molecule MS might further enlighten this aspect in the future.”

4. In the main text on page 4, the authors state that their mass spec analysis covers 27.4% of the expressed proteome, while on page 5, they state that their data (which data?) recapitulates (sic) 62% of known protein complexes, with 70% of their known components. Can these numbers be broken down to analyze what types of components (small? Large? Membrane-associated? Post-translationally modified?) are missing, and how these various percentages relate? What is the source of "known complexes" in this case? Is this the network analysis? (the statement "see above" is unclear) The authors have used a variety of sources, combined in various ways, to validate their communities, so a defined list or source for what the final validation dataset of "known complexes" is would be useful. Could the authors reference the precise workbook page in their supplementary material, if this is already provided there? If not, could it be added?

We agree with the reviewer that this part was not well explained and we clarified this point on page 6. In short, the benchmark includes all structurally characterized protein complexes found in the protein data bank (PDB) from any eukaryotic species plus all experimentally identified yeast complexes [PMID 20620961] – those were matched to our fractions. This list is available in Table S3, workbook page “PDB|AP-MS combined benchmark”. A prediction of membrane proteins with signal peptides is now included in Table S4, under workbook page “Protein Communities”, columns H and I, as predicted by HMMER (hmmer.org).

5. Which samples were used for negative stain and cryo-EM analysis? Were these cross-linked or not? If they were cross-linked, it would be useful to also include a micrograph of non-cross-linked fractions as a control.

The samples for negative-stain and cryo-EM analysis were fractions that were NOT treated with a cross-linker. For clarification, this information is now added in the Results section, page 7-8: *“To demonstrate that crude cellular fractions are amenable to the structural characterization of protein communities, we examined the different fractions for recurring structural signatures using EM (Fig. 4B to D) without adding any cross-linker for further stabilization of interactions.”*

6. A final note about the discussion of the EM analysis - the authors acknowledge that they specifically examined their fractions for "structural signatures", essentially searching for obvious, very large, easily identifiable stable complexes. While their ability to derive a de novo structure of ctFAS is admirable, the broader applicability of this to non-purified samples is doubtful, given that the majority of stable complexes will be significantly smaller. Some discussion of these types of problems would be useful.

We added the following sentences into the discussion section: *“The broader applicability of cryoEM to non-purified samples will be limited by the abundance and the stability of the protein communities during the lysis procedure. However, methods to improve the stability of these interactions, potentially with cross-linking prior to fractionation or lysis would allow discovery of further dynamic interactions and protein communities and would allow further simplification of the protein mixtures for structural study using this pipeline.”*

Minor points:

1. Figure S1 - header on panel B has the word "across" misspelled

We corrected this in the revised version:

2. Text page 5 (see note 3 above) - the word data is plural, so the "data recapitulate"

We corrected this in the revised version:

“Overall, the protein communities include 62% of the set of known protein complexes (the set of known PDB and AP-MS data, Table S2) with 90% average coverage of their components (Fig. 3 and Table S4).”

Reviewer #3:

The authors have separated lysates from the fungus, *Chaetomium*, into crude fractions, which they have analyzed in detail with mass spectrometry and cryoelectron microscopy to determine the abundance, interactions and structure of their constituent proteins and complexes. They present evidence that these constituents form dynamically interconnected "communities", which are often dissociated by more rigorous subcellular fractionation methods but have been preserved in these crude fractions. In a striking application of state-of-the art electron microscopy, the authors could identify and structurally characterize a single complex and present evidence for a previously structurally unobserved interacting partner. Despite this, there are a number of issues with the work as currently presented.

The idea of "protein communities" is not at all new, indeed it has been around since the inception of the interactome concept (which has thoroughly connected PPIs for which no stable complex has yet been observed), and it has been well understood for some time that our current "stable complex-centric" view of the interactome is incomplete (e.g. Menche, J. et al. Disease networks. Uncovering disease-disease relationships through the incomplete interactome. *Science* 347, 1257601 (2015)). Indeed, many of these concepts underlie the basics of subcellular fractionation (e.g. De Duve, C. Exploring cells with a centrifuge. *Science* 189, 186-94 (1975)). How does this "communities" concept differ from or significantly modify or contribute to the already well-established scale-free models of protein interactions in modern interactomics? (e.g. Barabási, A.-L. & Oltvai, Z. N. Network biology: understanding the cell's functional organization. *Nat Rev Genet* 5, 101-113 (2004)).

We agree with the reviewer that other groups (i.e. Kristensen et al. *Nat Methods* 2012 [PMID 22863883]) have used gentle cell fractionation for the characterization of protein complexes, and that the concept of protein communities has been proposed before.

We have incorporated the suggested citations into the manuscript (introduction, page 3). We want to stress though that in the absence of an experimental framework to capture protein communities, this higher level of organization has not been properly and generally characterized by previous studies.

While the EM work presented is extremely impressive, on page 4 it is claimed that a failure to observe aggregates by EM is evidence of a lack of non-specific / post-lysis effects on the protein interactions observed. This seems an unreasonable notion. Firstly, any non-specific interactions will be distributed across a continuum of possible states of complexation and structures (reflecting the lack of specificity and probable lack of physiology), and therefore will not be likely to present coherent features permitting the grouping and classification needed by the EM methods used. The claim that functional ontology is a secondary validation for this method, also seems to be misplaced - in a simple example, groups of nuclear proteins may be more likely to interact non-specifically with other nuclear proteins than cytoplasmic proteins, post-lysis, i.e. despite lack of any physiological significance. Or, any protein may be more likely to interact with related proteins, metabolically speaking, once the cell is broken, although those proteins may normally be spatially separated and never make a contact in vivo. Suffice it to say, while GO enrichment has its uses and can permit hypothesis formation regarding the potential relationship between groups of proteins (which would be an appropriate way to present the observed GO enrichment here), it does not constitute validation of specificity or physiology. The cross-linking results do not provide a more compelling argument either, since they will trap what is present, post lysis, at sufficient concentration - bona fide or not - and, overall coverage appears to be low (probably due to sample complexity).

First, we would like to thank the reviewer for crediting the EM work.

Second, we respectfully disagree with the comment “that a failure to observe aggregates by EM is evidence of a lack of non-specific / post-lysis effects on the protein interactions observed.” As a matter of fact, electron microscopic analysis is routinely used by the structural biology community to monitor the structural integrity and preservation. Even very low amounts of aggregates are obvious in electron micrographs and unfortunately way too often observed by structural biologists. We therefore think that it is very important to stress that they were absent from our fractions. In our view, this is a very important quality control, which is not yet routinely employed in the field of interaction proteomics.

Third, we also respectfully disagree with the reviewer’s general negativity about unspecific post lysis interactions. We believe that lysis is above everything a biochemical dilution step, which shifts the equilibrium of second order reactions towards dissociation. As such, and unfortunately, many weak but physiological interactions are lost during lysis, instead of being artificially created. As a matter of fact the vast majority of previous biochemical, structural and interaction proteomic studies have been relying on post-lysis methods (see e.g. PMIDs 26344197, 22863883, 24043423). We think it is an unreasonable notion to claim that all these studies have described artifacts.

Forth, we agree with the reviewer that functional ontology is not the best measure of validation. However, we did not claim that GO term enrichment is used as validation, but rather used this phrasing: “They are therefore probably functionally relevant as we observed that co-eluting complexes share the same functional ontology (independent two-sample t-test p-value = 3.88E-50, Fig. S5) or directly interact (XL-MS experiments, see below).” If we captured proteins with different GO term enrichment within the fractions, this would have been very concerning - that is why this analysis was done. Further, the calculation of a clear and coherent biological network (Fig. 3) backs up the identification of relevant interactions, especially by considering the high overlap with experimentally determined interactions.

Fifth, we agree that the coverage of the XL-MS data was relatively low in the original version of the manuscript, but we improved this by including additional experiments (please see also response to comment #2 of reviewer 1). In general, XL-MS will capture the more abundant residues that are spatially exposed to each other. Unspecific aggregates, if cross-linked are stoichiometrically diluted as compared to specific interactions and thus unlikely to be identified. They also would have been apparent during the benchmarking of the XL-MS data against known structures.

Overall, we appreciate the comment of the reviewer that, in general, during cellular homogenization procedures, artifacts could occur but we believe we undertook any conceivable effort to control for this and certainly more than any previous study. We have addressed these concerns in the discussion section. We have also down weighted the sentence regarding GO term enrichment by adding in the Results section “*They are therefore probably functionally relevant as we observed that co-eluting complexes share the same functional ontology (independent two-sample t-test p-value = 3.88E-50, Fig. S5) or directly interact (cross-linking experiments, see below), suggesting a functional relationship.*”

Fatty acid synthase also represents one of the most abundant complexes by far in yeasts, its subunits ranking ~20th most abundant (top 5%) of all proteins (<http://pax-db.org/> and references therein). It is not particularly surprising, then, that this could be identified and characterized in a crude fraction. This segment would have been much more impressive if a significant number of less abundant complexes (exosomes, exocysts, populations of different coated vesicles, etc.) had been separately identified and classified, at least at a primary level.

Abundance is indeed a limiting factor for the structural validation. The further fractionation hinted at in figure 4D shows that a wealth of structurally uncharacterized complexes could be solved with further fractionation without tagging or full purification, and correlating MS and EM but this is beyond the scope of the current study and is included as a demonstration. We already describe the aforementioned limitation, correctly pointed out by the reviewer in our discussion section. With our current methodology, only highly abundant protein complexes are accessible to cryo-electron microscopy and our integrative methods, which is limiting the structural validation but not as much the network analysis. However, with further advances in automation for cryo-electron microscopy both in hardware and software, access to less abundant structural signatures by extensive image

acquisition could be possible. We therefore believe that our work nicely exemplifies how more comprehensive structural proteomics studies might be designed in the future. We refer to this caveat in the discussion of the revised version of the manuscript, by adding: “*Further advances in EM acquisition and data analysis methods might further improve the coverage and identification of protein communities in the future.*”

We also want to stress the novel aspects of the FAS structure itself: it is (1) structure from a different organism of biotechnological interest; (2) resolved from a crude fraction instead from highly purified samples (3) the highest resolution of FAS achieved by cryo-EM, with additional, previously unobserved features, including a different catalytic intermediate in fatty acid synthesis.

The potential to support the metabolon hypothesis is exciting, as it is well known that supposed metabolic complexes are extremely difficult to co-purify despite knowledge of functional interactions and co-localization data by microscopy. The method presented most likely does not constitute one with innate qualities that improve the likelihood of observing metabolons from a sample preparation perspective, but rather, leveraged the sensitivity of the MS instruments used in conjunction with standard fractionation and cross-linking to find a nugget to weakly support the potential power of the method. Perhaps if the paper were organized in a metabolon-centric manner (with more examples) it would be clearer how substantial and significant the enrichment of otherwise difficult to capture, intact metabolic complexes is.

As mentioned above, we have improved the cross-linking analysis during the revisions and we now have validated many more of the interactions suggested by our network approach (please see revised Figure 3, updated text in Results section, page 7 and Table 1). We would prefer not to rewrite the paper in a metabolon centric manner, since many of the identified communities are not metabolons.

Much time is spent describing the need for in situ analytical means but the methods employed here are anything but, and it is unclear if these crude fractions really do present a major advantage over advanced subfractionation and purification methods using e.g. cross-linking and careful optimizations to preserve weakly-interacting partners, or true in situ high resolution cryoEM tomography.

If the authors want to make this claim they should devote time to rationalizing or at least explaining their protein extraction and preparation choices. Why that chosen extraction solution? For example, both EDTA and Mg⁺⁺ are used (and EDTA is listed twice) – why?

We apologize for these typos that we corrected.

Secondly, the handling time is significant compared to many modern approaches - 45 min of centrifugation (at a very high speed compared to typical extract preparation of ~20k RCF - probably to avoid slow protein concentration by ultrafiltration - and then the concentration step (how long?). By the time the sample is ready for loading on the SEC column or otherwise carried to the next experimental manipulation, a lifetime has passed (in terms of molecular half-lives in any random in vitro solution) - further making the claims of lack of non-specificity to be heavy-handed. In any case, the centrifugation used probably pellets stable, giant non-specific aggregates (which is indeed analytically important) - but one must ask about the accounting: what percentage of every protein complex highlighted in this study was found to be partitioned in the supernatant as opposed to the pellet? What is the partitioning bias of the communities described? At a bare minimum the authors should do some A/B/C comparisons of different protein extraction conditions, and/or extracts in the same condition prepared at different concentrations, in order to lend weight to their claims of what is happening in vitro and what observations are robust and reliable. Overlapping those data with established in vivo / in situ data (and ignoring high throughput IPs, or at least +/- such data) might reveal which condition is most useful for maintaining the in situ milieu, and therefore most suitable for accurate characterization by all the other methods employed.

We agree that our method is not “*in situ*” and we wanted to stress that it is complementary the upcoming *in situ* structural biology methods that visualize the cellular proteome but fall short of directly identifying its content. This is extensively described at several places in the manuscript (abstract, pages 3 and 10).

The protocols including lysis, extractions, centrifugation, etc are derived from our previous AP-MS studies in *S. cerevisiae* [PMID 16429126] and have been extensively and carefully optimized (and it is not in the scope of this manuscript to report these trial experiments). In addition, the biochemical

protocol (including the 100,000g centrifugation step) is similar to the one reported by the more modern approach of Kristensen et al 2012 [PMID 22863883].

Finally, and very significantly - there's simply no proper validation in this study. If the conclusions are valid - and probably some of them are - then it should be easy enough to select a hand full of nodes that represent interacting community members, and via deletion or mutation of those nodes, observe a phenotype and/or alter (subtly or significantly) the fractionation / MS profiles - this study has led the authors to hypothesize a particular behavior and arrangement of protein complexes/communities, and now they should test that claim by orthogonal experimentation to the path that led them there (and draw our attention to numerous interactions found in this study that resolve long standing paradoxes concerning functional associations that could not be shown to be physically linked previously). Which unexpected, un-described, or counter-intuitive nodes have been revealed here and contribute meaningfully to a physiological cascade they have not previously been (functionally) drawn into? Of course phenotype detection can be challenging - but at least some candidates must be validated in vivo to solidify the conclusions. I question that anything about this study can be characterized as in situ and recommend against that phrasing.

We have significantly expanded the cross-linking MS analysis to validate our protein communities (please see revised Figure 3).

2nd Editorial Decision

26 May 2017

The referees are now positive and we are pleased to inform you that we will be able to accept your paper for publication pending the following minor amendments:

- with regard to the notion of "protein community", it might be useful to include a "didactic Box" to explain the definition of "protein communities" and the approaches used to identify them experimentally and computationally. This might be useful in view of the recent study by Huttlin et al.
- we would need the ORCID for the following authors: Anne-Claude Gavin
- an author contributions statement should be added
- please submit individual figure files
- please include a form Data availability section. We would be grateful if you could follow the following general format:

"##Data and software availability

The primary datasets produced in this study are available in the following databases:

- RNA-Seq data: Gene Expression Omnibus GSE39462
<https://www.ncbi.nlm.nih.gov/geo/query/acc.cgi?acc=GSE39462>
- Proteomics data: PRIDE PXD000208 <http://www.ebi.ac.uk/pride/archive/projects/PXD000208>
- Quantitative growth measurements: Dryad Digital Repository doi:10.5061/dryad.35h8v
- [optional title/short description of the measurement type:] [full name of the resource] [accession number][resolvable hyperlink]
- List of protein complexes: Dataset EV1
- Results of benchmarking: Dataset EV2
- etc..."

- In the main text, please update call-outs from 'Fig S1' to 'Appendix Fig S1', etc. Also update accordingly the labels in Appendix file
- Please rename "Table S1-S5" to "Datasets EV1-EV5" and refer to these as such in the text.
- If you agree, we would suggest to shift all Materials & Methods in the main text, including the references.
- Perhaps it would also be a good idea to provide the full network shown in Fig S8-10 as Cytoscape file.

REFeree REPORTS

Reviewer #1:

The authors substantially revised their text and conducted additional experiments to address the points raised by the reviewers. My concerns have been addressed. I believe that the manuscript is now a strong candidate for publication and will be of interest to a broad readership.

Reviewer #2:

I am satisfied with the revisions made in response to my comments. I recommend publication of this outstanding piece of work.

2nd Revision - authors' response

12 June 2017

The authors made the requested changes and submitted the final version of their manuscript.

3rd Editorial Decision

20 June 2017

Thank you again for sending us your revised manuscript. We are now satisfied with the modifications made and I am pleased to inform you that your paper has been accepted for publication.

Corresponding Author Name: Dr. Anne-Claude Gavin; Dr. Martin Beck

Manuscript Number: MSB-16-7412R